

# ANYCaRE: A role-playing game to investigate crisis decision-making and communication challenges in weather-related hazards

Galateia Terti[1], Isabelle Ruin[1], Milan Kalas[2], Ilona Láng[3], Arnau Cangròs i Alonso[4], Tommaso Sabbatini[2] , Valerio Lorini[2]

[1]Univ. Grenoble Alpes, CNRS, IRD, Grenoble INP, IGE, F-38000 Grenoble, France
[2]KAJO, s.r.o., Slovakia
[3] Finnish Meteorological Institute, P.O. Box 503, FIN-00101 Helsinki, Finland
[4]Catalan Water Agency, ACA, 08036 Barcelona, Catalonia, Spain

*Correspondence to*: Galateia Terti (galateia.terti@univ-grenoble-alpes.fr)

**Abstract.** This study proposes a role-playing experiment to explore the value of modern impact-based weather forecasts on the decision-making process to i) issue warnings and manage the official emergency response under uncertainty and ii) communicate and trigger protective actions at different levels of the warning system across Europe. Here, flood or strong wind game simulations seek to represent to the players realistic uncertainties and dilemmas embedded in the real-time forecasting-warning processes. The game was first tested in two scientific workshops in Finland and France where European researchers, developers, forecasters and civil protection representatives played the simulations. Two other game sessions were organized afterwards i) with undergraduate University students in France and ii) with Finnish stakeholders involved in the management of hazardous weather emergencies. First results indicate that multi-model developments and crowdsourcing tools increase the level of confidence in the decision-making under pressure. We found that the role-playing approach facilitates interdisciplinary cooperation and argumentation on emergency response in a fun and interactive manner. ANYCaRE experiment was proposed, therefore, as a valuable learning tool to enhance participants' understanding of the complexities and challenges met by various actors in weather-related emergency management.

## 1    Introduction

Extreme weather and climate events challenge weather forecasting and emergency response operations and are often related to high social, environmental and economic impacts worldwide (WMO, 2015). In Europe, recent examples of devastating weather events include the 2003 European heat wave from which France suffered the worst losses with almost 15,000 deaths from August 1 through August 20 (Hémon and Jougla, 2004), the 15-16 June 2010 flash flood event in the Var Department



in France that caused the loss of 26 people (Poussin et al., 2015; Ruin et al., 2014), and the catastrophic fires in the forests of central Portugal that killed 64 people and destroyed more than 480 houses on 17 June 2017 (Mayer et al., 2017).

It is apparent that modern technological improvements including increases in accuracy and lead-time of the hydro-meteorological forecasts alone do not guarantee reduction of fatalities and economic disruptions (Petrucci et al., 2018; Terti et al., 2017a). Effective disaster risk management relies not only on the accuracy and precision of official hazard predictions and related warnings issued by forecasters but also on how those are communicated to and interpreted by end-users to support informed decision-making on allocating human and material resources before and during the crisis (Kox et al., 2018). When a severe weather phenomenon such as a big storm threats an area with flooding within a European country, forecasters in governmental forecasting offices use raw meteorological data and model outputs to run hydrological models and inform emergency services and other user groups such as road maintenance services for the imminent risk on life and property in the area of their responsibility. This information may be in the format of text messages or visual warnings and are transmitted to emergency managers in the control centres and are further disseminated to other public authorities (e.g., local fire stations) or voluntary organisations to prepare for action. Before deciding on any emergency actions, emergency managers examine the forecast information and deal with uncertainties seeking the most concrete indications for both the weather event and its potential impacts. In the words of the World Meteorological Organization (WMO), the emergency management decisions need to be supported by the knowledge not only of 'what a hazard will be' but rather of 'what a hazard will do' (WMO, 2015). In addition, the Sendai Framework for Disaster Risk Reduction (2015-2030) calls for an approach that is multi-hazard (i.e. takes into account the interaction of natural and man-made hazards), people-centred (i.e. takes into account the needs and rights of the affected persons) and preventive (i.e. aims at completely avoiding the potential adverse impacts of a disaster through action taken in advance) (Müller et al., 2017).

In this direction, recent decision-support tools promote the elaboration of multi-hazard 'impact-based' or 'risk-based' forecasts that translate meteorological and hydrological hazards and related cascading effects into sector- and location-specific impact estimations as the core to improve responders' and public's understanding and coping capacity to those risks (Luther et al., 2017). It is hypothesized that impact-based information systems integrating social vulnerability and behavioural processes in the forecast-warning system will help emergency services to better capture and respond to life-threatening situations and catastrophic scenes emerging from the conjunction of the hazard and social vulnerabilities that evolve in space and time (Creutin et al., 2013; Ruin et al., 2008; Terti et al., 2015, 2017b).

To take a first step towards exploring this hypothesis, we propose a new role-playing experiment that engages participants in the decision-making process at different levels of the weather-related emergency system (from hazard detection to citizen response). The focus of the experiment is a tabletop or pen-and-paper role-playing game (PnP) for adults in which



participants act their role through speech[1] while sitting in a comfortable setting (Cover, 2005). The PnP game is structured in progressive simulations in which improved multi-model outputs, including information on i) impact assessments and maps and ii) live data on exposure and vulnerability derived from social media and crowdsourcing (called "impact-based vulnerability information" hereafter), are presented as new decision-support tools to the players representing an emergency

group. The simulations are built based on the hypotheses that dynamic (near)-real-time impact information (e.g., potentially affected population and critical infrastructure, economic damages) can support emergency services to:

i. Locate spatially and temporally critical spots for intervention and therefore, better allocate available resources to protect lives and livelihoods.

ii. Communicate more targeted warnings and emergency guidance messages to help the public understanding how
certain hazards may affect their life, livelihood and property leading to appropriate self-preparedness and self-protective actions.

An important advantage of the simulation approach is its dynamic nature that allows participants to experiment "real time" decisions and experience potential changes in the outcome over time (Pasin and Giroux, 2011). The game design allows the player to progressively examine modern forecast data and impact-based vulnerability information and to interact with the

15 others and debate on relevant emergency activities. This "learning by doing" process - a fundamental principle in experiential learning theory (Kolb, 1984) - is privileged to take place in an informal setting without real consequences.

Role-playing games (RPGs) are the virtual simulation of real-world events especially designed to educate, inform and train the players for the purpose of solving a specific problem (Bowman, 2010; Drachen et al., 2009; Susi et al., 2007). In this study, ANYWHERE Crisis and Risk Experiment[2] (ANYCaRE) is developed with the aim to provide a wide range of

20 audiences with the opportunity to enhance their understanding of the recent uncertainties and dilemmas embedded in the real-time warning and emergency response processes. Through (semi-) realistic "what if" scenarios the players of ANYCaRE game are invited to assess different information describing an imminent risk situation and to decide collectively what protective actions, if any, are needed. Given the scarcity of chances to actually experience demands for decision-making on weather emergencies, playing the simulation-based game is a simple but essential mean for participants to nourish

their recognition of the emergency management difficulties and beneficial tactics (Crichton et al., 2000). On the other hand,

---

[1] Pen and paper or tables are not strictly necessary for the game. The terms pen-and-paper and tabletop are rather used to distinguish this format of role playing game (RPG) from other formats like Live Action Role-playing (LARP) in which participants act their characters physically as well.
[2] The acronym includes the name of "EnhANcing emergency management and response to extreme WeatHER and climate Events" (ANYWHERE) European Project (EC-HORIZON2020-PR700099-ANYWHERE) in which the experiment was developed. ANYWHERE project is an innovating action that aims at developing and implementing a pan-European decision-support platform integrating cutting-edge forecasting technology. More information about the project is available online at http://anywhere-h2020.eu.





playing may provide an excellent opportunity for current or future decision-makers to train on dynamic and uncertain incidents that may occur in times of weather-related crisis.

Although evaluating the effectiveness of role-playing or computer-based serious games is not always a straightforward task (Backlund and Hendrix, 2013), the literature values such educational or training games as motivating experiential learning

tools that go beyond traditional passive learning approaches often applied in conferences and seminars (Boyle et al., 2014; Dieleman and Huisingh, 2006; Salas et al., 2009). In the field of disaster risk management, role-playing has been successfully used to increase public awareness as well as to promote preparedness and prevention of losses (Rebolledo-Mendez et al., 2009). Examples of applications include role-playing simulations on flood management on cultural heritage for university students (Huyakorn et al., 2012), online serious games on natural disasters (e.g., tsunami, earthquakes) for

children (Pereira, Prada and Paiva 2014), and board serious games on resilience to geological hazardous events for school students or adult stakeholders (Mossoux et al., 2016). Effective designs of scenario-based crisis management game simulations have been also proposed to create opportunities for emergency trainees to rehearse crisis decision-making and prepare for real emergencies (Friman, 1991; Walker et al., 2011).

According to Bowman (Bowman, 2010), the role-playing method: i) 'enhances a group's sense of communal cohesiveness

by providing narrative enactment within a ritual framework', ii) 'encourages complex problem-solving and provides participants with the opportunity to learn an extensive array of skills through the enactment of scenarios', and iii) 'offers participants a safe place to enact alternate personas through a process known as identity alteration'. Therefore, it is hypothesized that the methodological approach adopted in ANYCaRE allows:

- Collaborative argumentation on weather crisis management (Huang et al., 2010). Dialectic reflection on weather
uncertainties and challenges helps participants to form their situational awareness and build a common strategy to solve problems of safety during extreme weather events. Therefore, the experiment facilitates collaboration and coordination between participants who may have distinct field of expertise and/or belong to different national or local institutions across Europe.

- Training of decision-making skills for emergency management (Linehan et al., 2009b). Through progressive simulations
the players are expected to get more and more familiar with good practices in emergency management. Serious games are recognized in the literature as useful tools for training since they offer an environment were trainees can experience demands of emergency management under stress before the real crisis (Crichton et al., 2000). The reception of new data as the game progresses makes the player to cultivate 'soft skills' such as communication and understanding of auxiliary or inconsistent information in limited time (Linehan et al., 2009a).

- Foster relevant behaviours for emergency response (Meesters and Van De Walle, 2013). A safe playing environment where participants act out given roles (sometimes very different from their duties in daily life) permits them to gain deeper understanding of the weather-related risks and decision-making complexities (Rebolledo-Mendez et al., 2009). During the experiment the player can realize conflicting requests arising in times of weather crisis and reconsider the relevance of specific (self-) protective actions.



Cognitive research indicates that in order for serious games to serve as effective educational means, they should embed the learning outcomes within the game mechanics and provide direct and specific feedback to the actions taken by the players (Bogost, 2007). The learning outcome of ANYCaRE is the improved weather risk-related decision-making for emergency response through modern multi-source inputs. Therefore, the game presents an environment where it is beneficial to

communicate with others to exchange complementary or contradictory information presented to the roles and consider competing demands before deciding. The learning outcome is embedded in the game play by providing specific feedback on how well the players are performing immediately after decision-making in the various states of the game. Thus, participants get informed for new forecast products and informational systems, and then practice their understanding and interpretation of those tools through playing. By observing players' debate and decision-making processes during the simulations the

experiment aims ultimately to obtain first conclusions on "if "and "how" improved multi-model (and potentially multi-hazard) outputs, including information on impact-based vulnerability data, can support the decision chain in European warning systems towards better responses.

The targeted audience for ANYCaRE varies from students and young researchers to developers, potential users and other stakeholders from different European agencies. The paper presents the first implementations of the experiment in the

emergency management of flood (and flash flood) and strong wind events. First, we describe the research methodology and the conceptual framework for weather-related crisis management adopted to design ANYCaRE. Section 2 explains also how ANYCaRE is set up and portrays general guidance for the design of role-playing experiments including simulations for weather-related crisis. Then, we provide details on the storylines and the roles selected for the flood and strong wind simulations, respectively. Section 4 presents the first applications of the game tested in different contexts. First results from

these experiments are reviewed based on the players' opinions on the gaming experience but also on the modern products provided as input data to support decision-making in the simulations, providing insights on related limitations. Finally, further advancements, extensions of ANYCaRE and opportunities for future applications are discussed.

## 2    Research methodology: Designing ANYCaRE

### 2.1    General concept and hypotheses

#### 2.1.1    Principles of tabletop role-playing games (TRPGs)

TRPGs is the original form of role-playing games and in principle they are conducted through discussion within an interactive and collaborative storytelling system (Aylett et al., 2008). The players act out the roles of characters and through the game they decide the actions of their character given certain rules and responsibilities within an hypothetical setting (Cover, 2010). In ANYCaRE game, participants are invited to play specific characters of the decision-making chain (i.e.,

forecasters, emergency managers or public) in an interactive storytelling related to a weather hazard in a European context. As a narrative-based game, ANYCaRE consists of a storyline and the simulations to be played by the group of roles. Table 1





presents a short description of the key game components and a question-based guidance for the primary design choices to be made. This table may serve as a design guide for future developers of educational or training exercises in the domain of natural hazards management.

The term game mainly refers to an array of simulations that represent certain real-event situations, which are played and manipulated by the participants. The players are guided through the simulations to experiment the outcomes of their decisions and learn from their playing experience (Dieleman and Huisingh, 2006; Huyakorn et al., 2012). A central function in TRPGs is the Game Master or Moderator (GM) who leads the storytelling during the game simulations (Heinsoo et al., 2008). The GM acts as organizer, arbitrator and moderator of the multiplayer role-playing game and therefore, it should be a person who knows the game in details and is able to answer questions regarding its rules. The game mastering responsibilities could be split among two or three people (e.g., designers or organizers of the experiment). The basic role of the GMs is the same in almost all traditional role-playing games, although differing rule sets make the specific duties of the game mastering unique to that system (Tychsen et al., 2007). For example, in ANYCaRE game the GMs is responsible, among others, to provide feedback to the players about the hydro-meteorological observations highlighting the relevant safety decisions that should have be taken at each playing round (similar to a weather reporter).

## 2.1.2 Conceptual framework for weather emergency management

The roles to be played and the potential decisions/actions to be chosen by the players in ANYCaRE are pre-defined based on qualitative evidence gathered during European workshops that took place in March and April 2017 (Müller et al., 2017) and in previous research (Ruin, 2007). Especially, the game was designed in order to be adapted or easily adaptable to most European countries' warning and emergency decision-making contexts. Based on examples of warning systems from Switzerland (Canton of Bern), Spain (Catalonia Region), Italy (Liguria Region), Finland (South Savo Region) and France, commonalities of these systems were identified and used to simulate realistically the dynamics of the warning and response processes starting with the detection of a potential weather-related threat and ending with decisions related to the coordination of the emergency response (Fig. 1). This process involves the identification of three major components: i) the type of actors involved in the warning system and their role in the decision-making process; ii) the timeline/temporality of the warning phases; iii) the types of actions/decisions the actors need to take in each phase to insure people's safety.

Depending on the country and its administrative and legal organization, different types of actors may be involved in the warning system. The responsibility for emergency management is also dependent on the spatial scale of the event. Most often, the municipality is the first level of responsibility and the most important actor of the emergency management as soon as the event does not exceed the capacity or the boundary of this local level. Mayors are in charge of the activation of municipal emergency plans. Nevertheless regional and national levels support their decisions with means of weather-related hazards forecasting and warning capabilities (Meteorological, Environmental or Hazard-related expert institutes) and regional emergency management centres and rescue services. In times of emergency, actors with complementary competencies are gathered (at the administrative level of concern: local, regional or national in case of State Emergency)



either physically or remotely to take decisions on how to best manage the crisis to insure people's safety. Generally, those Emergency Operation Centres (EOCs) include representatives of civil services as weather/hazard experts, police, fire fighters and rescue forces, representatives of municipalities, and infrastructure experts from public or private companies (road, telecommunication, energy suppliers). A representative of the highest authority concerned will act as the leader of the group

to organize the discussion and finalize the emergency decisions. The Centre's functions as the kernel of information, receiving, checking, sharing information with operational teams, deciding upon complex emergency actions that needs an holistic view of the situation and coordination efforts and communicating with the public.

To reflect this standard type of crisis management organisation, ANYCaRE game proposes to take decisions in the context of a simulated EOC gathering in the same room all or a choice of the actors cited above. A panel of roles' description to be

distributed among the players (randomly or based on their real-life expertise) describes the tasks and responsibilities of each player to contribute to the collective decision finalized by the group leader.

As shown in Figure 1, the timeline or temporality of the warning process often relates to 3 phases (Müller et al., 2017). The first one is a detection phase where model-based forecasts indicate the first signs of a potential high-impact event. At this early stage (3 to 10 days before the event), forecasts are neither accurate in space nor in time and the potential for impacts is

difficult to assess. The second phase starts when the event becomes more probable and the location and timing grow less uncertain. This phase, starting about 2 days before the event, relies on model-based forecast sometimes combined with some first empirical observations. At this stage, the first precautionary measures can be envisaged in order to prepare for the alert phase.  In the third phase, the alert one, when the magnitude of the event is almost certain as well as its location and timing, it is time for emergency actions. Decisions are mostly based on ground truth observations made by different operational

services involved in the emergency response. The temporality of those phases varies according to the type of hazardous events, as some are less predictable than others like in the case of flash flooding versus flooding events. Short-fuse events originating from convective storms for instance cannot be detected several days in advance which contracts the warning and response process in a very short period of time.

Based on these observations, ANYCaRE game addresses this temporal aspect of the decision-making process by proposing a

timeline adapted to the pace of the hazard under concern. Three rounds of decision-making are played successively to simulate the progression of the hazard from its early detection to its landfall. Nevertheless, based on the dynamic and predictability of the simulated event, the pace of succession of the hazard and/or risk information and decisions to be made in the game can vary to represent few hours to few days of the real life. Based on weather-related hazard information, the players simulating the EOC need to take decisions related to the three phases of the hazard progression described before. At

the beginning of the simulation, only weather and/or hydrological model-based forecasts are available for the coming hours or days. The level of uncertainty is still high. Round after rounds, more precise information including impact-based information is provided to reflect the decrease of uncertainty and the potential imminence of the event occurrence. With this information, specifically distributed to each role with respect to its own responsibilities, the players need first to interpret and share their specific knowledge before envisaging and deciding upon solutions to face the potential threat they identified.





Based on their collective evaluation of the situation, they have a certain time to choose between 3 types of decisions: i) stay aware and monitor the situation; ii) take actions in the context of a warning phase and activate the EOC to take precautionary measures; iii) activate the emergency plan and proceed to specific safety measures. Based on the selected emergency activity, the group further agrees if they would provide some generalized advice for safety to the public (e.g., "If inside,

5    move to higher floors", "Be prepared for electricity disruptions") or if they would proceed to more detailed emergency orders in specific area(s) of the territory (e.g., "Evacuate immediately"). To reflect the time pressure that real life EOC always faces, the time left to come up with a collective decision is limited for each round and the GM is in charge of pressing the group to obtain their decision in time. Since certainty in the warning-response processes is a key element of efficient crisis management, the group is asked to evaluate their confidence on the taken decision.

## 2.2    Experiment set up

### 2.2.1    Steps of the experiment

The experiment progression can be summarized in four progressive steps: i) introduction to ANYCaRE, ii) role assignment, iii) simulations, and iv) debriefing as proposed in Table 2. The experiment begins with the description of the setting by the

GMs. Then, each player is provided with a certain role defining his responsibilities during the game. Players get a few minutes to become familiar with their role and to introduce it to the rest of the group before the main game simulations start. Following the principles of tabletop role-playing, the GM facilitates the playing by presenting key aspects of the story and by triggering relevant discussion among the group of players. The game-designers' team acts as observers of the playing process and based on their observations they facilitate the post-experiment debriefing.

### 2.2.2    Objective of the simulation

In each simulation the EOC selects warning or emergency response activities based on the available information and related uncertainties. Following this first step, the players have to select (among some pre-established options) the best way to communicate their decisions to the targeted public. For example, forecasters need to interpret the hazard model outputs to choose the level of warning to be issued and communicated to the emergency managers and the general public. Then,

emergency managers evaluate the situation and decide what to do based on the forecasters' inputs and their own assessment of the level of exposure, potentially supported by impact-based and crowd-sourced information. Based on the discussion and advices from the others, the person designated to act as the leader of the EOC has to come up with a collective decision to be implemented and communicated. The role of the emergency management group is to keep the population safe and ensure smooth execution of everyday life activities in the territory while managing a given budget. In case of weather uncertainty,

this is a challenging task including a set of dilemmas. Deciding to alert the population and push people to stop their daily activity to take protective measures in areas not hit by a hazard might create unhappiness and loss of confidence in public





authorities. On the other hand, people are looking forward to recreational events such as big festivals; potential cancelation requires careful consideration to preserve people's wellness without risking their security. Every decision taken by the emergency management group has a consequence either in terms of human safety or wellness and economic value. The objective of the group is to undertake emergency activities from a pre-defined decision-reporting list avoiding actions that

might prove to be unnecessary at the end of the game. The proposed list of actions depends on the storyline and the purpose of playing, and can be easily adapted to different scenarios to be played from different audience. Thus, different versions of worksheets are created to fit the game implementation (see Appendix A).

### 2.2.3    Playing rules

A playing group in ANYCaRE should be preferably composed of 10 to 12 players. Collective decisions are recorded by

filling up one worksheet for the whole group in each game round. By using multiple rounds we allow the players to experience evolving hydro-meteorological facets and test different decision-support tools, which give more and more accurate information, as it gets closer to the event occurrence. Each round is composed of two trials, one where only existing basic hydro-meteorological forecasts are available and the second where additional more sophisticated decision-support products are provided. The group reports its choices in the relevant "TRIAL" column on the worksheet at the end of each

trial. To simulate time restrictions and pressures realized in real-world weather crisis decision-making, the players are given a limited time to provide responses in each trial. For example, in case of three round simulations, each trial lasts 10 minutes in the first two rounds whereas in the last round each trial lasts 8 minutes. Therefore, the first two rounds last 20 minutes each and the third round lasts 16 minutes (or less if the players are fast to take decisions on a given round). This time includes 1-2 minutes for the GM to present a short summary of the (hydro-) meteorological situation and consequences of

the decisions that have been taken at the previous round. The experiment should preferably last less than 2 hours (including the debriefing time). A detailed penalty scheme with "costs" in terms of safety, emergency resources, wellness etc. may be assigned to each activity listed in the worksheet or not. If the objective of the playing is the training and or information/education then specific penalties may be not necessary. Triggering debate and discussion to enhance understanding is a major priority. If penalties are assigned, the designer(s) should ensure that every action has a straight

consequence/penalty and that the initial credits given to the players are adequate even for the worst-case scenario.

### 3    ANYCaRE's Implementations

### 3.1    Flood Scenario

### 3.1.1    Storyline and roles

Inspired by European locations we introduce "Anywhere City"; an imaginary agglomeration including three distinct areas:

30   A, B and C, located on the slopes and at the foot of highlands drained by two fast-reaction rivers (Fig. 2a). Area A is



characterized by relatively steep tree-covered slopes drained by a small basin (e.g., 265 km$^2$) known for its fast response to precipitation. A suburb of about 1,000 residents and one school is settled on the slopes. There are no permanent settlements or critical infrastructure in flood prone zone but one campground located in the forest close to the riverbed (within the 10-year return period flood prone zone). Area B, is composed of both highlands and lowlands drained by a river basin of about

3,000 km$^2$. The densely populated urban area (e.g., 100,000 citizens) is located in the lower part of the basin. It includes the majority of schools, hospitals and other public services. About 30% of the residential, commercial areas and public services are located in the 20-year return period flood prone zone. Finally, area C is typical lowland with a large floodplain located in the lower part of a larger river basin (up to 4,000 km$^2$). There are no permanent settlements in C but the area surrounds the main bridge of Anywhere City, calibrated to resist a 50-year flood. The area is characterized by seasonal agricultural activity

and a recreation place where the annual festival of Anywhere City named "AnyDay" is taking place.

The game takes place at the beginning of fall and starts on a Monday, five days before the AnyDay Festival takes place with outside activities across the river and a big concert with famous singers close to the bridge area (area C in Fig. 2a). The peak of the festival is planned for Saturday when participants are expected to reach the number of 10,000. Public officials are checking out the weather forecasts to ensure that Anywhere City's traditional festival can happen in the best safety

conditions so that participants can enjoy next weekend camping and celebrating in the region.

Each player is given a specific sub-role to act as representative of one of the following institutions: i) hydro-meteorological services, to interpret the hazard model outputs and communicate warnings if needed; ii) first responder services to deal with possible evacuation of residences, schools, campsites and public events, iii) municipality, to make decisions related to the every day life (e.g., anticipation of school pick-up time, cancelation of school-related transport) or recreational events (i.e.,

AnyDay Festival) in the city; iv) road services, to manage road closures and the maintenance of the main bridge road in case of flood emergency (see worksheet for flood scenario in Appendix A). The highest challenge is to decide if AnyDay Festival should be cancelled or if the event could be maintained and probably set up flood protection measures in the bridge area. Withdrawing such a big event, for which people have prepared and have been looking forward to, obviously would cause upheaval and would reduce their wellness. In this case, the municipality would also pay cancelation fees and other expenses

with the ultimate goal to prevent people from risk. On the other hand, if high water or debris flow blocks and collapses the bridge while the festival is still on, thousands of unprepared people will be exposed to severe flooding. In general, if no protective action or evacuation were decided for the day an event hit an area, the population is considered subject to risk.

### 3.1.2 Input data and simulations

The period of concern in flood simulations are the five days of the week (from Monday to Friday) preceding the day of

"AnyDay Festival". The GM provides and comments medium-range deterministic precipitation forecasts and hydrological forecasts produced by the European Centre for Medium-Range Weather Forecasts (ECMWF) for Monday and Tuesday in order to familiarize the players with the products and slowly put them in the context. Each of the three following days represents one round of the game for which collective decisions are requested from the players, assuming that the group



decides every evening for the emergency activities of the next day (Fig. 3a). In each round, the players receive area-specific information so that they could make distinct safety choices adapted to the predicted hazard in each area of Anywhere City. Low-predictability events such as flash flooding in steep slopes and domino effects like debris flow in the river are also part of the flood scenario to prompt emergent stressful situations.

In the first trial of each round, players receive the 24-hourly accumulated precipitation based on the ECMWF ensemble mean for Anywhere City. Precipitation is calculated as the median value in every gird cell from 51 integrations with approximately 32-km spatial resolution. Secondly, input data include hydrological forecasts for 3 different locations (i.e., A, B, C) based on the European Flood Awareness System (EFAS), which is operationally running at the European scale under the Copernicus Emergency Management Services (Copernicus EMS) (Smith et al., 2016b; Thielen et al., 2008). EFAS

products were particularly selected because they present successful implementation of probabilistic medium range flood forecasts with special attention on the communication of uncertainty (Demeritt et al., 2013; Pappenberger et al., 2013) and their operational availability EU-wide (Smith et al., 2016a).

The second trial proposes additional radar-based high-resolution rainfall accumulation forecasts (i.e., every 15 minutes with lead times up to 6 hours) produced by the European Rainfall-InduCed Hazard Assessment (ERICHA), and EFAS bias-

corrected forecast hydrographs where near real-time discharge observations are used to compensate the propagation of the error between forecasted and observed discharge (Bogner and Kalas, 2008). Advanced hydro-meteorological products are selected to show the importance of end-user tailored forecast visualizations and highlight the potentiality of combining i) continental scale medium range forecast in coarse spatio-temporal resolution with ii) local, high resolution and short range predictions to improve timely detection and better localization of extreme events. These include return-levels hydrographs

(i.e., time-series of the return levels of corresponding forecasted discharge values at the same time-step) and exceedance plots (i.e., daily box-plots corresponding to different lead-times of the hydrological forecast)[3] from EFAS forecasts driven by the ECMWF ensemble.

The second group of products presented in the second trial focuses on the importance of translating the hazard into risk information, and especially probabilistic risk assessments, for the decision-making. Impact-based products are based on the

Rapid Impact Assessment layer[4] that combines event-based hazard maps with exposure information to assess several categories of impacts such as affected population and damages (Dottori et al., 2017). Direct economic losses are computed combining the 'Coordination of information on the environment' (Corine) map with flood hazard variables (i.e., flood extent and depths) and a set of damage functions derived for European countries. In addition to that, the extension of urban and agricultural areas affected is computed using the Corine Land Cover (Dottori et al., 2017). Last information presented to

---

[3] Every box contains a value representing the probability of exceeding certain critical threshold in that day (e.g. 50% probability of exceeding 5-years return level).
[4] The Rapid Impact Assessment Layer is demonstrated online in the ANYWHERE's catalog at http://anywhere-h2020.eu/catalogue/?product=river-flood-impact-forecast.





players, emphasize the benefits of the seamless integration of the volunteer geographic information (VGI) collected during the crisis from social media and via dedicated crowd-sourcing applications (i.e., artificial Tweets and posts in a hypothetical crowdsourcing system especially drawn for experiment). VGI content is being widely recognized as crucial source of information that cannot be measured directly (e.g., socio-economic impacts, human perception) and thus, it is assumed to

positively contribute to the overall situational awareness and crisis decision-making.

### 3.2    Strong wind Scenario

#### 3.2.1    Storyline and roles

The strong wind scenario was built based on the structure of the flood scenario with necessary adjustments. The experiment was originally designed for a combined training day of Finish civil protection, meteorologists and electricity company

employees in South Savo municipality. Therefore, the territory and the game roles were selected as representative of the study area. South-Savo municipality has a special landscape with i) dense forest areas, ii) powerlines above the ground that are vulnerable to strong winds and falling trees, iii) lake areas attracting numerous boatmen during summer, and iv) the urban areas of Mikkeli and Juva. In the storyline, we illustrate areas A, B and C to represent the variety of these different areas (Fig. 2b). Area A represents the urban area of Mikkeli with a year-round-population of 50,000 persons. The game takes

place in mid-July and starts on Friday, 24 hours prior the Jurassic Rock Festival and Sulkavan Soutu Boatrace. The music festival is expected to attract additional 25,000 persons to Mikkeli (Area A) and 4,000 boatmen at the boat race on the lake of Sulkava (Area B). The urban area includes also many critical infrastructures like chemical factories, hospitals and shops that are dependent on continuous electricity supply. Finally, Area C mainly contents the road leading from Juva to Sulkava, which is a crucial evacuation route for civil protection when facing emergencies at the lake areas. Since the road tends to get

blocked by falling trees in storm situations, it was included into the game storyline to create an additional challenge for the civil protection's decisions.

The players of the strong wind scenario act as: i) meteorologists, to interpret the NWP model information to the customers and issue warnings (and the corresponding meteorological bulletin); ii) civil protection, to make decisions for possible evacuation of public events (e.g., Jurassic Rock Festival); iii) electricity company, to manage the maintenance of electricity

distribution to the customers considering the related economic constrains (see worksheet for strong wind scenario in Appendix A). The electricity company representatives are especially challenged to divide their resources effectively in order to fix potential power cuts as quickly as possible and ensure the electricity supply in the area. Similarly to the flood scenario, the biggest challenge for the players refers to the protection of the festival planned for Saturday.

#### 3.2.2    Input data and simulations

The timeline in the strong wind simulations is 24 hours before the public events. Given the rapid development tendency, the small size and the short lifecycle of convective storms, the forecast time is among the shortest lead times in weather





phenomena making forecasting a real challenge. Therefore, each round is chosen to be in terms of hours prior to the event (i.e., Round 1 – 24h, Round 2 – 6h, Round 3 – 1h ahead) in ANYCaRE game (Fig. 3b). From meteorological perspective, the scenario imitates the so-called 'Asta' strong wind event that happened in 2010 and caused financial losses of over 20 million euros (Astola et al., 2014). This particular event was selected because most of the trainees for whom the experiment was

designed had indeed experienced Asta strong wind. Living through the event again (but without knowing it at the beginning of the game) and facing it under different circumstances, the players were offered the possibility to compare the existing tools to the new impact tools, and to realize the possible consequences of such a strong wind event in different situations (e.g., occurrence during daytime on the busiest summer weekend compared to the original event that took place during night).

The simulations replicate the KRIVAT[5] briefing where the different actors hold a teleconference on Fridays to receive a weather outlook for the weekend. In the first round (24h prior to the event), the GM provides information about the synoptic situation and the meteorological forecast of ECMWF and the Global Forecast System (GFS) in the area. The scenario presents also imaginary but realistic cascading effects and unpredictable events such as falling trees over the power lines and subsequently, power outages in critical areas (e.g., close to the chemical factory). In such urgent situations, the civil

protection needs to react fast to cooperate and communicate with the meteorologists and the electricity company's staff. In each round, existing tools with only weather information are presented in the first trial whereas the second trial provides additional impact-based tools. In particular, in the second trial meteorologists receive model calculations outputs from the "Expert System for Consequence Analysis and Preparing for Emergencies" (ESCAPE) model calculations, and interpret them for the use of civil protection. The ESCAPE-model has been designed for evaluating the dispersion of toxic gases into

the atmosphere and the effects on humans and the environment (Kukkonen et al., 2017). Second-trial tools give emphasis on the identification of the severity of the convective cells, their detailed route and the economic impact estimation that is particularly interesting to the electricity company. A common approach for the automatic identification and short-term forecasting of thunder storms is the object-oriented storm tracking, which follows the movement of individual storms from remote sensing data and extrapolates the storms based on the tracking information (Rossi et al., 2013). Severity categories of

convective cells are classified by using failure data from the electricity company and compose an interesting input for forecasters and civil protection. This information is assumed to facilitate also the optimization of electricity grid actions undertaken by the electricity company. Also, in the second trial civil protection has access on the A4FINN[6] platform and the

---

[5] KRIVAT is a platform supporting collaboration between enterprises and authorities in major disruptions in services. The service is aimed for critical infrastructure organisations. More information available at: https://www.erillisverkot.fi/en/services

[6] A4FINN is a map product developed for the municipality of South Savo. A4FINN is combining weather information with impact estimations. It combines data which used to be scattered to many different information sources. For instance, the emergency response plans of the civil protection are now connected with weather information which brings the experienced





available impact map. The platform provides a tool indicating the weather situation severity level in different municipalities. A4FINN includes relevant weather parameters and a map product, which shows the automatic, suggested preparedness level of the civil protection.

## 4    ANYCaRE Testing and Applications

The first ANYCaRE experiment was carried out in the frame of a European workshop in Helsinki (Finland) on September 2017[7]. This workshop gathered 161 attendees including researchers, forecasters, civil protection and representatives of related companies in Europe. A game session was organized with a group of sixteen players to compose the virtual EOC of Anywhere City and play the flood scenario (Fig. 4a). The main purpose of the experiment was to check the validity of the scenario and test the functionality of the game with experts from different fields and agencies across Europe (Table 3). Another flood game testing was organized four months later in a European scientific meeting in Grenoble (France) with similar goal to confirm improvements and enhance the game mastering experience before further applications. Among participants in the two test experiments there were PhD students and researchers in weather-related hazards, developers and modellers, emergency managers and operational forecasters (Fig. 4b). Therefore, players' experience in weather-induced risk modelling and management ranged from low (i.e., < 5 years) or no operational experience to high (i.e., > 10 years) providing a genuine diversity to be reflected in the games.

After the simulations, the game observers invited the players to provide feedback on both the experiment set up and the input data presented in the game. The test experiments were considered successfully since the game was found 'to be representative of the reality of flood crisis management' by players who had the experience of such situation and 'clearly demonstrated the benefit of certain products'. In both Helsinki and Grenoble Experiments, players agreed that the simulations were motivating and 'kept the stress', and the whole set up was a 'very representative group exercise'. That made participants to appreciate the game and recommend it for play in future events. Players introduced further potentials for ANYCaRE emphasizing:

- Training scopes such as coaching emergency services in order to sharpen their emergency agility and alertness before the crisis strikes: 'The game can be used in my organization to test the emergency response' [Helsinki player]; 'I would like to use the game as part of the training in the hospital emergency dissipation. Good way to improve their way of taking decisions' [Grenoble player].

---

and less experienced emergency responders to the same level when making decisions. More information available at: http://anywhere-h2020.eu/wp-content/uploads/docs/D1.3.pdf.

[7] The experiment took place on September 21, 2017 within ANYWHERE's 2nd workshop in Helsinki where 90 project partners belonging to the consortium and 71 from external organizations participated to follow up ANYWHERE's innovations and contribution to the response-to-weather-extremes era.





- Educational contexts: 'It is a very useful exercise and I would love for it to become also a tool to educate different segments of the population' [Grenoble player].

Considering such potentialities, two more experiments were organized in spring 2018 as presented in Table 4. In the first experiment, ten undergraduate students of the L3 Pro[8] study program of the University of Grenoble Alpes (UGA) played

ANYCaRE flood simulations as part of their course (Fig. 4c). Students following this scientific and professional training are especially prepared for a practical integration into the operational world of water resources and risk management. The benefit of this gaming practice was evaluated through a short questionnaire answered both before and after the game play.

The second experiment was integrated in the official annual training of the Finnish stakeholders involved in the management of hazardous weather emergencies (i.e., meteorological services/FMI, civil protection in the Eastern Finland region,

electricity providers from "Järvi-Suomen Energia"), which was held on May 2018 in Mikkeli (Finland) (Fig. 4d). Impact products developed by the Finnish Meteorological Institute (FMI) were introduced to the trainees with the aim to familiarize end-users with new developments, introduce them with the products' beneficial use as operational decision-support tools, and rehearse their communication towards effective response during weather hazards like severe strong winds. A post-experiment questionnaire was also designed especially to assess if the game met the learning objectives as expected.

Based on improvements proposed by participants in the test experiments, the game designers considered various ways to ease the gaming process in the two new game sittings such as i) documenting and distributing the input data for every role in separated booklets; ii) supplementing the initial worksheet with more specific assignments for certain roles (e.g., road closing to be decided by the road services representatives in the flood scenario); iii) distributing demanding roles according to the real-life competencies and players' background (e.g., give forecasting responsibilities to players with forecast

experience). To trigger debate rather than winning spirit in the performed experiments, no specific penalties were assigned to the options listed in the worksheets. Instead, in the first round the GMs highlighted the three-fold common goal of the emergency managers to: i) insure citizens' safety and prevent loss of life; ii) prevent disturbances in social life, which make people unhappy and reduce wellness; iii) minimize their expenses for protective measures compared to the actual needs.

## 25  5   First results and Discussion

The four experiments included a total of 46 players. All of them participated in the debriefing phase orally, and optionally by shortly recording their main ideas on paper. Participants commented on either positive remarks or suggestions for improvements related to the game itself but also to the modern products provided as input data to support decision-making in the simulations. Short textual annotations on post-it notes were gathered and were placed in the corresponding category on a

board to open the discussion (Fig. 4e). Table 4 presents a summary of the themes in which the feedbacks are classified after

---

[8] Equivalent to the 3rd year of a 3–year Bachelor of Technology program.



qualitative analysis of the players' written comments (Rebolledo-Mendez et al., 2009). In the 3rd Experiment, the students replied also to short pre- and post-experiment questionnaires. Similarly to previous studies on role-playing gaming (Huyakorn et al., 2012), the questionnaires were composed by the same open questions to identify differences (if any) in their perceptions on risk management after playing ANYCaRE game. Trainees of the 4th Experiment had the opportunity to

provide also detailed feedback on the gaming experience and the related learning outcomes by answering Likert-scale questions as well as complementary open questions supplementing their statements.

## 5.1  Feedbacks from observations of the game-pay and debriefing discussions

### 5.1.1  ANYCaRE is a valuable communication and learning tool

It is important for participants to feel that the game experience was playful, the rules were understandable and the learning procedure was attractive (Dieleman and Huisingh, 2006; Turkay and Adinolf, 2012). After playing ANYCaRE game, the majority of participants in the four conducted experiments expressed their satisfaction for the gaming experience. Players noted their appreciation for the 'stimulating' (6 instances), 'very fun' (4 instances) learning approach experienced in ANYCaRE. It was mentioned that it was 'very interesting to try roles different from the usual day-by-day experience'

[Helsinki player]. The experiments were found to fulfil the main hypothesis that the game offers a safe environment to enact different personals and gain deeper understanding on the decision-making challenges met by the involved actors (8 instances): 'the game is a good way to start handling emergency management with a different approach. Also people that are not in their own role can understand the struggles of the others' [Grenoble player]; 'New experience that helps to understand better other roles' [Mikkeli player]; 'Good way to put you in the shoes of actors' [UGA player].

In all the experiments, the players recognized ANYCaRE scenarios as very realistic and presented a strong commitment to the storytelling (6 instances): 'Very real feeling and I really took the game seriously' [Grenoble player]; 'Very good game and incredibly realistic!' [Mikkeli player]. During the feedback sessions in Helsinki and Grenoble Experiments as well as in the Mikkeli Training, ANYCaRE players discussed that role-playing was a good approach to bring people from different institutions together to share knowledge and experience while examining new (pre-) operational tools and their potentialities.

They often expressed their preference on ANYCaRE's role-playing structure for training decision-making skills on emergency management (4 instances): 'Would use this exercise in our later trainings for civil protection' [Mikkeli player]. On the other hand, UGA students appreciated the game for educating them on their future responsibilities: 'Good exercise/learning for the following because it gives concrete example'.

### 5.1.2  Modern impact-based information increase the level of confidence in emergency management decisions

At the beginning of each round the GMs described the existing weather-related conditions and explained if and why the group decision taken in the previous round was relevant or not to ensure safety with the minimum losses. It appeared that the



players in the test experiments as well as in Mikkeli Training did a relatively good job already from Trial 1 in the first round (e.g., by evacuating scouts from the campsite before the occurrence of a sudden severe flash flood in area A in the flood scenario). Some players in Grenoble Experiment mentioned that improved forecasts including probabilistic information for exceeding return levels were very informative (4 instances). Though, in some experiments it was observed that advanced

meteorological forecasts such as high-resolution precipitation maps and flood probabilistic forecasts provided in Trial 2 did not necessarily lead to better decisions for emergency response. Still, this result may be subject to specific skills of the players who acted as forecasters as well as the overall experience of the participants in these sessions. For instance, operational meteorologists participating in Mikkeli training were able to detect significant signs about a potentially severe strong wind situation already from the first round, even with limited amount of information. Thus, they gave a very good

briefing to the others creating a confident atmosphere for the rest of the roles to select emergency actions. This was not the case when the much less experienced UGA students played the flood simulations. Although, differences between Trial 1 and Trial 2 were not always obvious in terms of the selected emergency activity on the worksheets, in most of the cases, players in the four experiments rated their confidence higher when they passed to Trial 2.

In the debriefing, some players of the flood scenario simulations mentioned that detailed meteorological data are not trivial

to non forecasting-specialists (4 instances): 'Difficult information for operators in emergency centres' [Helsinki player]; 'For a non-expert, it is difficult to understand the probabilistic info provided by forecasting systems' [Grenoble player]. Yet, hydro-meteorological forecasting was observed to dominate the decision-making and related discussion compared to impact estimations in the first two experiments. The limited use of the new impact-based tools in the game may be attributed to the lack of players' familiarity with the new products opposed to previously seen operational tools. Other potential explanations

that require further examination may include: i) inadequate understanding on how to handle the new impact information during crisis; ii) absence of trust in the new developments; iii) incapability of the adopted visualizations to convey helpful information. In UGA experiment, where students were much less experienced with the provided hydro-meteorological products, decisions between trial 1 and 2 changed highly considering the impact-based vulnerability information and especially, the social media posts.

In the debriefing discussions, the players agreed that the information provided especially from impact estimations and social media was very useful to better target their actions. Generally, in all the experiments players seemed to largely rely on impact observations assumed as reported through comments and pictures on social media. This was particularly the case in the third and last round of the three flood-scenario experiments where the players changed totally their emergency decision after receiving a crowdsourced image showing the bridge blockage with wood and debris. The payers shifted from "no

action" to the set up of flood protection measures in area C, and finally, the closure and maintenance of the bridge road in Trial 2. The examples of modern impact-based vulnerability information included in the experiments were found to reduce the overall uncertainty in the decision-making process. The multi-model outputs were characterised as useful especially when accessible to multiple actors directly without the need from mediation from other (governmental) agencies (3 instances): 'The ability to offer and share valuable data or information directly to consumer is brilliant. It avoids putting



someone's safety in the hands of government decision-makers' [Grenoble player]. Expert forecasters and members of the civil protection in Mikkeli Experiment found that the variety of information presented through the A4FINN tool and impact-based maps helped them to shape a holistic view of the situation and increased their confidence in decision-making. The players of all experiments acknowledged that the variety of new products enhanced their sureness about specific emergency

activities to be chosen and communicated in particular areas.

## 5.2    Analysis of the post-experiment questionnaires responses

In the Grenoble experiment with UGA students, the 10 game participants were asked the three same questions before and

after playing the game to evaluate the potential change of their perception related to weather-related crisis management. The open questions were the following:

- What key words do the terms 'crisis management' evoke for you?
- What types of information are necessary to take a decision with respect to an imminent inundation threat?
- According to you, what challenges crisis decision-makers have to face when flooding occurs?

The qualitative analysis of the differences between the answers pre- and post game indicates that students were not imagining how many exchanges and knowledge sharing, communication and coordination efforts were needed for crisis management. 'Anticipation', 'information', 'communication', and 'actors' are key words that were not cited before the game but did get cited several times after the game. With respect to the second question related to the type of information necessary for crisis decision-making, playing the game did also make a difference. Students realized that not only historical

data about peak discharges, damages or the identification of elements at stake were useful but weather prediction, vulnerability and accessibility data as well as more contextual information (like the ones concerning the occurrence of big events potentially increasing temporarily the exposed population) were also essential. Finally, in terms of the perception of the challenges faced by crisis decision-makers, the players highlighted the difficulty to manage the multitude of data, the prioritization of actions in order to better anticipate and choose the optimal decisions without being overwhelmed. Key

words like 'optimal decision/management', 'elements at stake' and 'anticipation' were mostly cited after playing the game.

After Mikkeli Experiment, five of the twelve participants also provided written responses to a questionnaire, which was targeted to "evaluate" the usefulness of the ANYCaRE gaming simulations in the training process. This low response rate could be attributed to the fact that everyone first participated to the oral debriefing session and did not feel the need to add any more comments. The questionnaire asked players to rate their satisfaction in terms of training experience and

applicability of their learning on a five-level Likert scale. In agreement with the oral debriefing discussion, most of the players who answered the questionnaire (3 out of 5) denoted that ANYCaRE fairly fulfilled their expectations highlighting that the simulations promote co-operation and communication between crucial actors toward the formation of common situational awareness and preparedness strategies. According to the players, forming the complete situational picture is one of the main challenges in weather crisis management and it was also depicted as a challenge while playing. Players evaluated





the learning outcomes of ANYCaRE though various questions. Nearly all respondents stated that they learned a lot through playing. All of the respondents highly rated their ability to apply this learning in their professional environment and they largely proposed the gaming activity as a relevant training tool. In summary, the respondents indicated the following take away messages:

• The training through the game was very pleasant. The process revealed the importance of certain roles in the decision-making chain and taught the significance of co-operation between multiple actors for efficient problem solving.

• The lessons can be immediately applied in the corresponding agencies increasing preparedness for different weather-related situations. The simulations help to get an overview of the multi-faceted demands in emergency response and to identify products that can be used for the anticipation of impacts in affected areas in the future.

• The experiment proposes a flexible framework that can be easily adapted to host the needs and purposes of different training activities (e.g., training of the municipality lead group to weather-related hazardous events). With certain modifications, ANYCaRE can be extended to other tabletop exercises or to be digitalized to offer a modern multi-task and multi-role structure for deeper understanding of decision-makers' challenges in weather crisis.

## 5.3 ANYCaRe's limitations

The biggest challenge identified in the role-playing exercise is to engage players equally and empower their active presence in the decision-making. In our experiments and similarly to previous game studies, the personality of the player was observed to be an important factor during the game; with the more extrovert and talkative players to dominate the decision-making and the shyer ones to participate much less in the debate (Mossoux et al., 2016). Shy players were involved in the
20 argumentation though, when they felt the need to defend a specific priority related to their role. In addition to that, players already experienced with certain processes and/or tools (e.g., experienced forecasters in Mikklei Training) tended to largely rely on their prior expertise when leading decisions in the game. This may hinder the reflection of certain facts especially tested in the experiment (e.g., confidence on the decision-making based on the presented information). To facilitate the representation and relevant intervention of all the roles and their particularities in the debate, it was suggested that very
detailed responsibilities and constrains should be assigned to each role at the beginning of the game; allowing the players to set up a common strategy with the other persons playing the same role (8 instances). Smaller groups with up to 12 players were also perceived as more controllable and efficient to prompt player's engagement in the game world.

The gaming process was diagnosed as having also limitations related to the time needed for the players to understand the concept and get ready to play (4 instances) as well as processing of complex hydro-meteorological information probably
unfamiliar to some participants (7 instances). Although the current level of complexity in ANYCaRE made the game to be stimulating for the players, it was observed that the given hydro-meteorological information were sometimes difficult to digest in a short time for participants without forecasting expertise. Obviously, more time should be given in the presentation and explanation of the elaborated data before the simulation starts. Another perspective is to introduce Level-1



simulations in which the forecasters' group will work separately to prepare and deliver forecasts to the emergency group (5 instances).

Timing is overall an important issue questioning the balance between having adequate time for debate / decision-making and also representing realistic stressful crisis situations. Although the time available for taking and recording decisions on the worksheets was increased after some propositions made in the test experiments, players in Mikkeli and UGA Experiments still found the ten minutes very limited, especially in the first round. However, designers should consider that as the total time required for the experiment increases, the applicability of the game in different settings (e.g., workshops, short training sessions) is hindered. A well-trained game master is definitely necessary to clearly introduce material appropriate to the players and guide them through the game, optimizing the total time required for the experiment.

## 6    Conclusions

This paper presents a role-playing experiment designed to investigate crisis decision-making in weather-related risks. ANYCaRE allows us to explore how decision-makers and stakeholders interact with scientific and operational outputs to better anticipate and respond to extreme and high-impact weather and climate events in Europe. The experiment includes tabletop gaming simulations on common hazards under consideration in European sites such as i) flooding and flash flooding, and ii) strong winds and thunderstorms. From September 2017 to May 2018 flood-based ANYCaRE was applied in two European Experiments and one educational experiment in a French University whereas the strong wind scenario was played for training purposes with Finish operational forecasters, emergency managers and stakeholders.

First results show that ANYCaRE game aroused the interest and enthusiasm of participants and offered to the players a protected environment to try-out emergency actions without facing true risk for human life. In both scenarios, the players affirmed that the simulations adequately reflected situations found in the real world and facilitated their involvement in the storytelling. It was observed that this participatory technique set a playful and collaborative atmosphere between scientific partners and stakeholders generating fruitful debate on appropriate emergency decisions. Players highlighted the value of communication between the involved actors, which was successfully represented in the game.  Compared to conventional trainings, the gaming approach presents the possibility to introduce more than one tool at a time to the end-users generating cross-interest to each other's decision-support tools and related challenges in the decision-making. Therefore, the gaming simulations sufficiently served the educational and training goals of the experiments.

As a proof of concept, this first study did not include yet an explicit evaluation of the products presented as inputs and their contribution to the decision-making. At the final debriefing step of the experiments, participants were rather encouraged to exchange knowledge, thoughts and insights on their capability or difficulty to decide and communicate their action based on the available information and given constrains. In a broad view, in all the experiments the players complied well with the scenario requests. The main conclusion drawn from the first applications is that modern multi-model outputs provided extra confidence in the decisions taken in the first trials based solely on existing hydro-meteorological information.  Expert




forecasters and members of the civil protection in Mikkeli Experiment found that the variety of information presented through impact-based tools helped them to shape a holistic view of the situation and increased their confidence in decision-making. Though, this fact was not necessarily obvious when examining the group responses on the worksheets. It appears that decision-making is largely influenced by personal attributes such as prior experience on weather forecasting and

familiarity with disastrous events. Concerning the vulnerability information, all the players of ANYCaRE games agreed that online crowdsourcing tools might be a great provider of ground facts necessary to enhance situational awareness of authorities especially in cases of high hydro-meteorological uncertainties and forecasting failures. Future developments were suggested to geolocate the social media content to help emergency responders to clearly identify places where urgent action is needed. Nevertheless, interviews conducted with emergency communication experts in the context of the ANYWHERE

project also show that the reliability of such crowd sourcing information is a concern and there is a necessity to check in real-time the information and potentially counter false rumour as soon as they emerge (Müller et al., 2017).

Communication to the public was only a small part of the players' duties in ANYCaRE's first implementations. To simplify the game the players did not have to compose an emergency message by themselves but they rather got some simplified examples to choose from. Therefore, choosing the relevant official emergency response and deciding on whether that should

be communicated to the population in specific locations of the study area was considered as the main responsibility of the emergency group. When the hypothetical crowdsourcing system indicated the bridge blockage in Anywhere City, for example, the emergency group of flood experiments commanded to inform the public immediately for the imminent risk. The players opted for specific guidance to the public indicating to stay away from the bridge and the festival area. Other communication aspects could be addressed in the future by asking players to choose specific format and delivery mean for

the conveyed information. That would enable us to examine what are their criteria for the distribution of understandable and usable messages to the public.

Rather than a single tabletop role-playing, we vision ANYCaRE as a broad experiment campaign that will encompass various versions of games to be applied in different settings. Adjustments of the current flood game are already considered to prepare experiments for Italian pupils (e.g., between the age of seven to thirteen years old) with the objective of raising

awareness on i) the crisis decision-making process and the challenge of school-related decisions, and ii) the appropriate behaviours of students and their parents in case of flash flooding. Applications to other weather-induced risks such as wildfires were also encouraged by participants in the past experiments and are discussed for implementation. A series of future expansions is considered to: i) adjust scenarios to other weather hazards including multi-hazard cases and complex cascading effects commonly challenging European cities; ii) test additional models and technological innovations (e.g.,

crowdsourcing tools, dialog systems, internet-based apps); iii) establish other formats of serious gaming such as online or board games to attract different audiences (e.g., stakeholders, general public, pupils) and subsequently, enlarge the amount and variety of feedbacks on weather crisis decision-making. Online playing would allow the player to experience multiple simulations until establishing a deeper understanding on relevant interpretation of the new information and successful paradigms for emergency response. Such developments will offer a floor for testing hypothesis on the usefulness of new





products and their reception by experts or the public. ANYCaRE digitalized or online playing could be used as a communication tool between developers of new decision-support tools and end-users to provide a feedback loop for further improvements to developers. More targeted questions could be examined: "Is the product user-friendly and easily understood?"; "Does it make decision-making easier for the users?"; "If yes, what is the characteristic that makes it useful

5    (e.g., the spatial or temporal specificity, precision)?"; "If not, then what needs to be improved?". When the game refers to the general public more detailed questions dedicated on risk communication could be addressed: "Does the population at risk receive the risk/warning information and emergency messages through the selected mean(s)?"; "Are the timing and geographic specificity of the messages appropriate to interrupt the public's daily routine activities in favour of crisis-related preparedness and protection?". Our ultimate purpose is to draw more detailed conclusions on the effectiveness of impact-

10   based forecast visualizations and delivered warning and emergency messages (i.e., content, structure and format) in terms of comprehension and mobilization of action. Such knowledge is prerequisite for the anticipation of effective crisis communication strategies and relevant emergency responses to prevailing weather threats.

## 7    Aknowledgments

This project has received funding from the European Union's Horizon 2020 research and innovation programme (H2020-DRS-1-2015) under grant agreement No 700099.

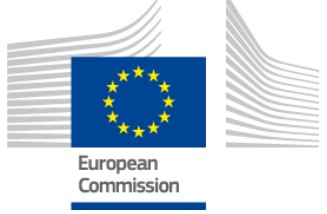

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





**Figure 1: Conceptual framework of the Weather-related European Warning Systems adopted for the design of ANYCaRE role-playing experiment. National, regional or local actors are framed within light, medium or dark grey rectangles, respectively. According to the warning/response phase in which they mainly operate, actors and their warning or emergency actions/decisions (snip single corner rectangles) are presented inside the light (detection), medium (hazard warning) or dark green boxes (emergency response). The dashed arrows illustrate the flow of information among national, regional and local actors. Adapted after** (Müller et al., 2017)**.**





**Figure 2: Presentation and brief description of the territory considered in the storyline of ANYCaRE for the: (a) Flood scenario and (b) Strong wind scenario. Each area of the territories includes attributes for special consideration in the emergency decision-making (e.g., camping, schools, dangerous intersections, industries) representing critical points for intervention in European cities.**



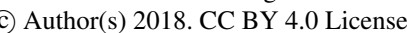

**Figure 3: Schematic illustration of the gaming timeline (rounds 1 to 3) and the information provided to the players of ANYCaRE for the: (a) Flood scenario and (b) Strong wind scenario. Each of the three game rounds (R1, R2, R3) played in the experiments corresponds to a daily or hourly time step before the festivals that, according to the storylines, are held on Saturday. In the second trial of each round, the players receive additional decision-support tools including high-resolution forecasts and impact-based vulnerability inputs.**



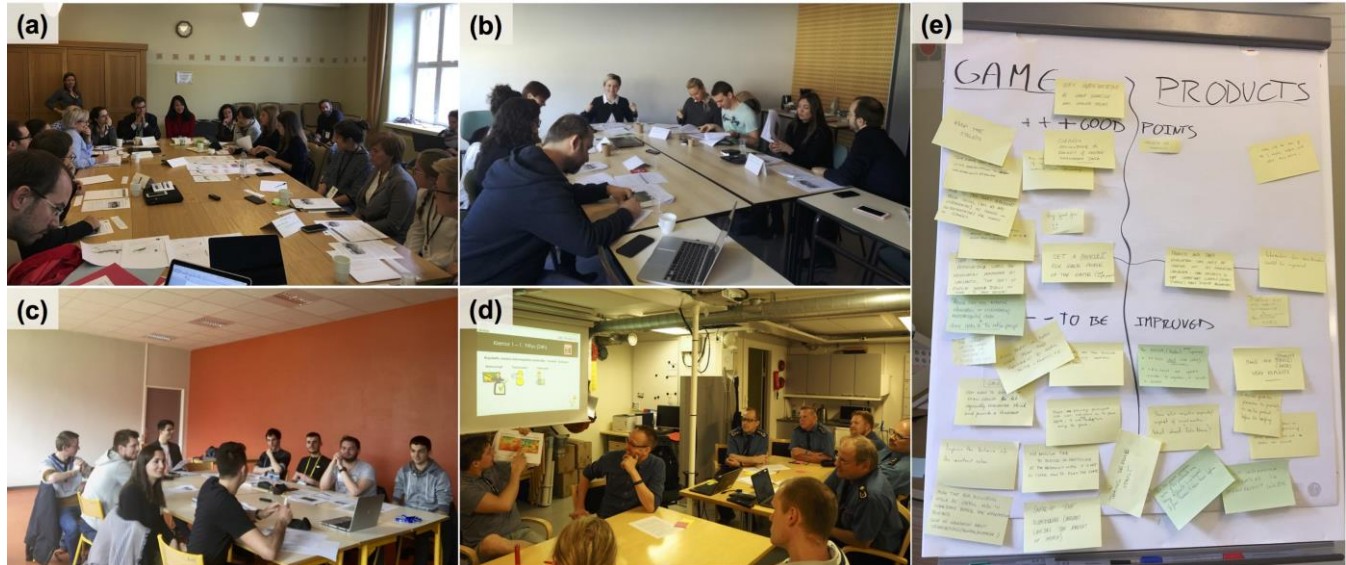

**Figure 4: From (a) to (d): Players' debate during the role-playing simulations of ANYCaRE in: (a) Helsinki Experiment (21 September, 2017); (b) Grenoble Experiment (24 January, 2018); (c) UGA Experiment (11 April, 2018); (d) Mikkeli Experiment (17 May, 2018). (e) Example of players' post-experiment debriefing post-it notes**
5 **(photo taken in Helsinki Experiment).**



**Table 1 Description of the game components and the corresponding design choices to be made when developing ANYCaRE.**

| Game components | General description | Design choices | Playing concerns |
|---|---|---|---|
| 1. Storyline | Territory and main attributes in which the story evolves | • Real world **territory** versus virtual one? Characterize the territory in terms of population, buildings and infrastructure, geographical location and topography (e.g., mountains, rivers, plains). <br><br> • What is the **hazard(s)** that will threat the territory? <br><br> • What elements may be **affected from the hazard**? List the special groups of population (e.g., campers, residents, parents) or special infrastructure (e.g., schools, power plants etc.) existing in the territory that may be impacted in certain ways. <br><br> • What is the **main interest** in the game? Define a realistic story (e.g., a festival to happen in the territory, a critical infrastructure like food or water system that is important for the population needs) that introduces some complexity and challenge in the decision-making. | • An existing territory may be used if we want to inform/train participants for a certain area. However, sometimes it is better that the players do not know the area in advance so that we do not insert bias and players are not influenced by past experiences. Also, with virtual territories it might be easier to combine different characteristics to address several issues related to different locations in the real world. <br><br> • In any case, a realistic main interest should be selected to make the game interesting to the players. |
| 2. Roles | Actors that will be represented in the storyline | • What is the **level of the decision chain** to simulate? Define if representatives of one or more of the 3 levels are needed in the story (Level 1: Weather Forecasters; Level 2: Emergency managers/civil protection; Level 3: General public and targeted users like private companies). | • Very detailed responsibilities and constrains are suggested to be assigned to each role at the beginning of the game allowing the players to set |



| | | • What **sub-roles** will be attributed to the players (e.g., expert hydrologist, mayor, first responder)? Define the specific interests/targets of each sub-role in the game. | up a common strategy with the other persons playing the same role. |
|---|---|---|---|
| 3. Simulations | Weather-related scenario, decision-making and timeline in which certain data are provided. | • What is the **(hydro-) meteorological scenario** to simulate? Define how the hazard will evolve and what will be the final outcome in the game based on a real past event or hypothetical scenario. • What is **the time frame and pace of the decision-making**? Select the season of the year the simulations refer to. How many days/hours will be simulated during the 1h of the game play? This choice depends on the speed of development and the forecasting capability related to the hazard under concern. The choice of the pace of the rounds of decision-making should reflect the real characteristic of the simulated event (e.g. few days or hours for short-fuse events, several days for slow developing events). • What are the **collective or individual decisions** to be made? List the corresponding emergency activities (e.g., evacuation of a certain area or infrastructure) and communication activities (e.g., issue warnings, provide advice messages, order evacuation) that the roles can | • Hypothetical scenarios are also presented during the storytelling with data extracted from past events. The difference with the real event data is that we do not need data to come from a specific year and location but we just combine one or more data sources to describe our storyline and scenario in a relevant way. • The emergency and communication activities will compose the worksheet that the player will fill at the end of each round of the gaming simulation. The worksheet should include all the concerns that we want to discuss in the experiment. • Every decision taken by players should have a consequence in the game. The designer may assign specific penalties to the options listed in the |





apply in the storyline. Every proposed action should correspond well with the responsibilities of the roles as well as the elements included in the storyline (e.g., specific locations, critical infrastructure and population in the territory).

- What will be the **input products**?
List the main products (e.g. forecast outputs, social media info, impact matrix…) that will be provided distinctly to the role players according to their area of responsibility. If the objective is to evaluate new products, "basic" products that are routinely used for decision making need to be provided in TRIAL 1 in order to compare with the new products given in TRIAL 2.

worksheet to challenge the decision-making. Even if there is no penalty scheme, each decision should conceptually have a "cost" (e.g., in terms of economic value, human safety or wellness). The objective of the players is to undertake emergency and communication activities avoiding "costly" decisions that might prove to be unnecessary at the end of the game.

- Input products are not necessarily operational. Representative figures that illustrate the input information are adequate.





**Table 2 Progressive steps of ANYCaRE experiment. The proposed durations suggest an experiment that lasts up to two hours.**

| Action | Potential Means | Proposed Duration |
|---|---|---|
| 1) The GM briefs about the procedure to be followed for the game and describes the storyline to the players. The rules of the game and the purpose of playing are clearly explained. | Power point presentation and talk | 12 minutes |
| 2) The players answer the short pre-experiment questionnaire *(optional - depending on the objectives of playing)*. | Questionnaire(s) printed in paper | 8 minutes |
| 3) The game organizers distribute the roles *(usually pre-selected based on the participants)*. | Written/printed labels | 1-2 minutes |
| 4) The players read the specificities of their role (3 min) and present their responsibilities to the rest of the roles (5 min). | Printed booklet with role description and input data for each role | 8 minutes |
| 5) The GM introduces the simulations by presenting the current meteorological facts and first forecasting. | Power point presentation and talk | 3 minutes |
| 6) The players play the simulations. The GM facilitates the playing by leading the storytelling and triggering relevant discussion among the group. In each round the GM ensures that the players respect the rules of playing and they fill and submit the decision worksheet timely. | Worksheet printed in paper | 56-60 minutes (about 20 min in each round) |
| 7) The players answer the short post-experiment questionnaire *(optional - depending on the objectives of playing)*. | Questionnaire(s) printed in paper | 8 minutes |
| 8) The observers/facilitators trigger the debriefing discussion. | Power point presentation of starting questions  Post-it stickers on a board | 15-20 minutes |



**Table 3 Information for the game sessions conducted within ANYCaRE.**

| Experiment / Location | Date | Implementation | Main Purposes | Number of Players | Players' Profile |
|---|---|---|---|---|---|
| 1st / Helsinki (Finland) | 21/09/2017 | Flood scenario | • Test the game functionality<br>• Identify needs for improvement<br>• Collect feedbacks on the usability of the experiment and potentialities for future applications | 16 | PhD students Researchers Developers Emergency managers Operational forecasters |
| 2nd / Grenoble (France) | 24/01/2018 | Flood scenario | • Test the game functionality after the first round of improvements<br>• Frame the usability of the experiment in specific future events | 8 | PhD students Researchers Developers Emergency managers Operational forecasters |
| 3rd / UGA (Grenoble, France) | 11/04/2018 | Flood scenario | • Increase students' awareness on given weather-related risks and relevant responses<br>• Educate future professionals and potential policymakers on crisis management challenges and | 10 | University students |



| | | | | | |
|---|---|---|---|---|---|
| | | | • decision-making processes | | |
| 4th / Mikkeli (Finland) | 17/05/2018 | Strong wind scenario | • Introduce new impact tools to end-users<br>• Collect feedback on new developments<br>• Train experts/end-users on crisis decision-making using new informational tools | 12 | Operational forecasters<br>Civil protection<br>Electricity company |



**Table 4 Themes of the players' comments recorded in the post-it notes in the debriefing phase of the four ANYCaRE experiments. Every comment was placed in one theme. Post-it notes with more than one thematic annotation were assigned to more themes to correspond comment-to-theme.**

| Positive comments on ANYCaRE | Instances | Improvements proposed for ANYCaRE | Instances |
|---|---|---|---|
| The game enhances understanding of emergency decision-making actors, processes & related challenges | 8 | Provide role-specific information to increase engagement | 8 |
| The game is stimulating | 6 | Provide more explanations on the hydro-meteorological input information | 7 |
| The game is realistic | 6 | Provide more time for decision-making, especially in the first round | 7 |
| The game is useful as a training tool | 4 | Create a different pre-game decision-making session for forecasters | 5 |
| The game is fun/ice-breaking exercise | 4 | Provide more information before the game starts | 4 |
| The game is useful as an education mean | 2 | Distribute demanding forecasting roles according to the real-life expertise | 4 |
| The game demonstrates the benefit of modern impact-based vulnerability information | 1 | Give the players penalty scores/limited resources | 3 |
| The game is useful as an evaluation tool for new products | 1 | Provide the input information concentrated in a booklet | 2 |
| It is interesting to enact a different role | 1 | Keep the worksheet short & simple (more time dedicated to the discussion) | 2 |
| The game facilitates collaborative argumentation and coordination | 1 | Add more roles and/or actions | 1 |
| The game is well structured for its purpose | 1 | Keep the number of players small | 1 |
| | | Add more surprising/ triggering elements along the scenario | 1 |
| | | Provide feedback to the players after every round | 1 |
| | | Add other hazard scenarios and domino | 1 |

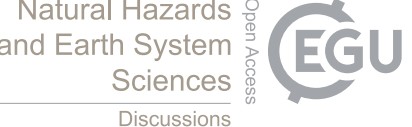



| Positive comments on the modern products (input data in the simulations) | Instances | Improvements proposed for the modern products | Instances |
|---|---|---|---|
| | | effects (e.g., fires) | |
| Total number of instances | 35 | Total number of instances | 47 |
| Improved (sometimes probabilistic) hydrological forecasts are very informative | 4 | Hydro-meteorological products are not trivial for non-forecasters | 4 |
| The multi-model outputs are useful and are delivered directly to the end-users under concern | 3 | Hydro-meteorological products should present additional parameters | 4 |
| The products are clear & understandable to the end-user | 2 | Better presentation of the crowdsourced info/social media | 3 |
| Impact-based vulnerability information (especially from social media) are useful | 1 | Need for more detailed input data | 1 |
| The development is relevant for operational use | 1 | Some models need further development | 1 |
| | | Schematic illustrations should be translated in the spoken language (not in English) | 1 |
| Total number of instances | 11 | Total number of instances | 14 |



**APPENDIX A**

**Table A. 1 Worksheet with pre-defined list of emergency and communication activities delivered to players of Flood scenario.**





| Flood Scenario / Worksheet for Emergency activities to be considered for the following day: | TRIAL 1 | TRIAL 2 |
|---|---|---|
| 1) No action is performed. You just follow the weather updates and keep monitoring the situation. | | |
| 2) Activate the Emergency Operation Center to coordinate the rescue services and operational forces. You may choose to take one or more of the following precautionary measures:<br><br>a. Set up flood protection measures in: area A    area B    area C<br><br>b. Anticipate or delay school pick-up time in: area A    area B<br><br>c. Cancel the school and the school-related transportation in: area A    area B<br><br>d. Close the main roads in: area A    area B    area C<br><br>*(Please circle the area(s) of your choice or indicate it in the Trial)*<br><br>e. Cancel the festival planned for the weekend in area C. | | |
| 3) Activate the emergency plan to trigger evacuation of exposed areas and vulnerable populations. You may choose to take one or more of the following emergency measures:<br><br>a. Evacuate campsite in area A.<br><br>b. Evacuate schools or shelter pupils in schools in area B.<br><br>c. Evacuate the festival in area C (if it's the day of the party).<br><br>d. Evacuate and shelter the population in: area A    area B    area C<br>*(Please circle the area(s) of your choice or indicate it in the Trial)*<br><br>e. Clean/maintain the bridge in area C.<br><br>f. Close the bridge road in area C. | | |
| 4) Deactivate the emergency measures for: area A    area B    area C<br>*(Please circle the area(s) of your choice or indicate it in the Trial)* | | |
| **Please rank your confidence on your decisions from 1 (no confident) to 5 (very confident):** | | |
| **Flood Scenario / Worksheet for Communication activities:** | TRIAL 1 | TRIAL 2 |



| | | |
|---|---|---|
| **1) No information to communicate.** | | |
| **2) Notify and forewarn about fake news and rumors shared in social media.** | | |
| **3) Provide general advice for safety.** As examples: "Ask for information before travelling" / "Do not drive into flooded areas. If floodwaters rise around your car, abandon the car and move to higher ground if you can do so safely."<br><br>    **area A**        **area B**        **area C** | | |
| **4) Inform the public for the emergency plan and communicate the decisions taken in "Emergency activities" card (choices in B and C).** For example, order schools directions or the residents in general to evacuate. "Evacuate immediately. Be sure to lock your home as you leave. If you have time, disconnect utilities and appliances. Return home only when authorities indicate it is safe."<br><br>    **area A**        **area B**        **area C**<br><br>(Please circle the area(s) of your choice or indicate it in the Trial) | | |
| **Please rank your confidence on your decisions from 1 (no confident) to 5 (very confident):** | | |

**Table A. 2 Worksheet with pre-defined list of emergency and communication activities delivered to players of Wind scenario.**

| Wind Scenario / Worksheet for Emergency, electricity companies and meteorologist's activities to be considered for the following: 24h/6h/1h time period | TRIAL 1 | TRIAL 2 |
|---|---|---|
| **Meteorologists' decisions (other roles can comment):** | | |
| **1) Issue the LUOVA-bulletin** *(Please circle your choice)*<br><br>   **Yes**       **No** | | |
| **2) Issue the DANGER-bulletin** *(Please circle your choice)*<br><br>   **Yes**       **No** | | |
| **3) Alert level of LUOVA-bulletin?** *(Please circle one color level)*<br><br>  🟢  🟡  🟠  🔴 | | |




| Civil protection's decisions (other roles can comment): | | |
|---|---|---|
| 4) **No action is performed. You just follow the weather updates and the possible LUOVA-bulletin's monitoring.** | | |
| 5) **Activate the Emergency Operation Center to coordinate the rescue services and operational forces.** *You may choose to take one or more of the following precautionary measures:*<br><br>a. **Cancel the Jurassic Rock festival in area A.**<br><br>b. **Cancel the Sulkavan Soudut boatrace in area C.** | | |
| 6) **Activate the emergency plan to trigger evacuation of exposed areas and vulnerable populations.** *You may choose to take one or more of the following emergency measures:*<br><br>a. **Evacuate/stop the camping in the area A.**<br><br>b. **Evacuate/stop the whole festival in the area A.**<br><br>c. **Evacuate/stop the boat event in area B.**<br><br>d. **Evacuate the population in:** *(Please circle the area(s) of your choice)*<br>  **area A          area B          area C**<br><br>e. **Are your resources sufficient?** *(circle one)*<br>  **Yes                No                Maybe**<br><br>f. **Alert level of the civil protection?** *(circle one)*<br>  🟢 🟡 🟠 🔴 | | |
| The electricity company's decisions (other roles can comment): | | |
| 7) **Activate the Emergency Operation Center for coordinating the forces.** *(circle your choice)*<br><br>  **Yes          No** | | |
| 8) **How many repairing teams will be sufficient amount to prepare for the situation?** *(circle one)(circle one)*<br><br>  **1  2  3  4  5  more** | | |
| 9) **Are your resources sufficient?** *(circle one)*<br><br>  **Yes                No                Maybe** | | |
| 10) **Alert level of the civil protection?** *(circle one)*<br>  🟢 🟡 🟠 🔴 | | |



| | TRIAL 1 | TRIAL 2 |
|---|---|---|
| **11) The coordination of the road clearance (fallen trees):** | | |
|     a. **Do you need clearance actions on some of the areas?** *(Please circle one or more areas and decide who's responsible)*<br><br>    **Area A    Area B    Area C**<br>    b. **Responsible institution?**<br>    **Area A:   Electricity company/Civil protection**<br>    **Area B:   Electricity company/Civil protection**<br>    **Area C:   Electricity company/Civil protection** | | |
| **12) Deactivate the emergency measures for** *(Please circle the area(s) of your choice)*:<br><br>    **area A      area B      area C** | | |
| **Please rank your confidence on your decisions from 1 (not confident) to 5 (very confident):** | | |

| **Wind Scenario / Communication activities:** | TRIAL 1 | TRIAL 2 |
|---|---|---|
| **1) No information to communicate.** | | |
| **2) Inform the public of the hazard**: *(Please circle the area(s) of your choice)*<br><br>    **area A      area B      area C** | | |
| **3) Inform the customers (electricity) about the situation:** *(Please circle the area(s) of your choice)*<br><br>    **area A      area B      area C** | | |
| **4) Inform the media:** *(Please circle your choice)*<br><br>    **Yes      No** | | |
| **5) Inform the public of the emergency plan and communicate the decisions taken in "Emergency activities":** *(please circle the area(s) of your choice)*<br><br>    **area A      area B      area C** | | |
| **Please rank your confidence on your decisions from 1 (not confident) to 5 (very confident):** | | |