# Peer review of "ANYCaRE: A role-playing game to investigate crisis decision-making and communication challenges in weather-related hazards"

_Natural Hazards and Earth System Sciences, 2018_

## Referee Comment (RC1) · Anonymous Referee #1 · 31 Oct 2018

***Details points:***

*Page 3:*
L5: *"The simulations are built based on the hypotheses that dynamic (near)-real-time impact information."* What is the meaning of *near* here?

L9: *"*Communicate more targeted warnings and emergency guidance messages to help the public understanding..." Specify the type of aimed audience.

L13: *"*The game design allows the player..." How?

L15: *"*Through (semi-) realistic "what if" scenarios..." What is the meaning of *semi* here?

*Page 5:*
L3: *"*The learning outcome of ANYCaRE is the improved weather risk-related decision-making for emergency response through modern multi-source inputs." Explain what is a *modern multi-source inputs*; give some examples.

L9: "By observing players' debate and decision-making processes during the simulations the experiment aims ultimately to obtain first conclusions on "if "and "how" improved multi-model (and potentially multi-hazard) outputs, including information on impact-based vulnerability data, can support the decision chain in European warning systems towards better responses." Decompose the sentence and give more explanation on this point.

*Page 6:*
L1: Table 1 requires more explanation in the text.

L16: *"*For example, in ANYCaRE game the GMs is responsible, among others, to provide feedback to the players about the hydro-meteorological observations highlighting the relevant safety decisions that should have be taken at each playing round (similar to a weather reporter)." What is the basis for the information that the GM proposes?

*Page 7:*
L9: *"*A panel of roles' description to be distributed among the players (randomly or based on their real-life expertise)". This choice of role is important and could be a part of the training/rising awareness strategy; it would be interesting to discuss this point in the paper.

L25: *"*A Three rounds of decision-making are played successively to simulate the progression of the hazard from its early detection to its landfall." Is it always 3 rounds (whatever the application)? Can you justify this choice?

*Page 8:*
L3: *"*Based on the selected emergency activity, the group further agrees if they would provide some generalized advice for safety to the public (e.g., "If inside, move to higher floors", "Be prepared for electricity disruptions") or if they would proceed to more detailed emergency orders in specific area(s) of the territory (e.g., "Evacuate immediately")." Can you give more details on this process?

L15: *"*Players get a few minutes to become familiar with their role and to introduce it to the rest of the group before the main game simulations start." Is it sufficient (notably for players without expertise in the field)?

L22: *"*Following this first step, the players have to select (among some pre-established options) the best way to communicate their decisions to the targeted public." How many options? What freedom is left to the players? Can they suggest new options?

*Page 9:*
*L12: "*Each round is composed of two trials, one where only existing basic hydro-meteorological forecasts are available and the second where additional more sophisticated decision-support products are provided." The interest of these two trials is only explained later in the text. You should introduce the justification of this game design choice here.

*L18: "*This time includes 1-2 minutes for the GM to present a short summary of the (hydro-) meteorological situation and consequences of 20 the decisions that have been taken at the previous round." How are evaluated/calculated these consequences?

*Page 11:*
*L1: "*In each round, the players receive area-specific information so that they could make distinct safety choices adapted to the predicted hazard in each area of Anywhere City." Why using different areas? What is the impact on the game?

*Page 13:*
*L17: "*In particular, in the second trial meteorologists receive model calculations outputs from the "Expert System for Consequence Analysis and Preparing for Emergencies" (ESCAPE) model calculations, and interpret them for the use of civil protection." Are these results pre-calculated or are there live simulations made according to the players' choice? In this case, what is the duration of a simulation?

*Page 15:*
*L17:* "To trigger debate rather than winning spirit in the performed experiments, no specific penalties were assigned to the options listed in the worksheets. Instead, in the first round the GMs highlighted the three-fold common goal of the emergency managers to: i) insure citizens' safety and prevent loss of life; ii) prevent disturbances in social life, which make people unhappy and reduce wellness; iii) minimize their expenses for protective measures compared to the actual needs." It is not clear. Can you reformulate/explain this point?

*Page 16:*
*L12:* "Players noted their appreciation for the 'stimulating' (6 instances), 'very fun' (4 instances) learning approach experienced in ANYCaRE." On how many instance? Was there also criticism from the players? Why did you choose this type of evaluation rather than an anonymous questionnaire?

*Page 17:*
*L12:* "Although, differences between Trial 1 and Trial 2 were not always obvious in terms of the selected emergency activity on the worksheets, in most of the cases, players in the four experiments rated their confidence higher when they passed to Trial 2." Trials 1 and 2 are combined in the same game session. Does that not introduce a bias in the comparison of the two trials? Shouldn't it have been better to make two independent experiments, one with classical information (i.e. trial 1) and the other with advanced information (trial 2)?
The variability between the groups are not discussed; it would be an interesting point.

*Page 19:*
*L4:* "In summary, the respondents indicated the following take away messages…" It would have been interesting to compare learning through play with that obtained with traditional methods (e.g. courses).

L23: "To facilitate the representation and relevant intervention of all the roles and their particularities in the debate, it was suggested that very detailed responsibilities and constrains should be assigned to each role at the beginning of the game; allowing the players to set up a common strategy with the other persons playing the same role (8 instances)." Another way to avoid the 'leader effect' is to give each player only part of the information, forcing them to communicate to build a common strategy.

*Page 21:*
*L23:* "Adjustments of the current flood game are already considered to prepare experiments for Italian pupils (e.g., between the age of seven to thirteen years old) with the objective of raising awareness on i) the crisis decision-making process and the challenge of school-related decisions, and ii) the appropriate behaviours of students and their parents in case of flash flooding." How?

---

## Referee Comment (RC2) · Piangiamore (Referee) · 14 Jan 2019

I think this paper is very interesting and is an important addition to the literature. It well describe the "ANYCaRE" role-playing game to simulate crisis decision-making and management arousing curiosity and making me want to play and deepen the subject deal with. After reviewing it, I would like to experience it myself as a player tester. The authors are able to explain and justify the game design choices in a detailed and punctual way rich in useful particular to understand the consequences of one decision rather than another on crisis management. ANYCaRE is presented as a serious games particularly useful to educate future crisis managers, as well as to spread a cul-

ture of risk also for general public promoting awareness and correct behaviors which are fundamental for public safety. A very precious tool for all (from citizens with no knowledge of the subject to experts). I believe the importance of this paper stems from the applicability of the approach to the several, but I have some reservations about a so difference players' scientific background. My suggestions are essentially that authors should also calibrate the game to inexperienced users, simplifying in this case the data sets (considered by someone too difficult to interpret). The adequacy of the development of the project described, as it is currently stands, shows the authors' analysis is both rigorous and generalizable. A very well written interesting manuscript, pleasant to read and fluent: a beautiful synthesis of a role-playing game with the education intent our Society needs that can be broadly and usefully applied.

---

## Author Comment (AC1) · 25 Feb 2019

We wish to thank both reviewers for taking the time to provide insightful comments and help us improve our manuscript. Please find below, our responses (R#) to each of the comments made by reviewer 1. You will also find as a separate file the pdf of the revised version of the paper with the track of changes.

Comments from Reviewer #1 The paper proposes a role-playing game to simulate crisis decision-making and management. The paper is particularly interesting; however, some points have to be improved to be eligible for publication. I would suggest the authors to consider the following points:

[Figure]

C1 : The paper lacks of a real state of the art on role-playing game dedicated to crisis management.

R1: With respect to the state of the art, we did our best to conduct a proper literature review on the subject, nevertheless the subject is vast and contributions may come from disciplinary fields that we are not yet familiar with. To fill this gap we added a couple of paragraphs (p5-6) to complete the literature review especially with respect to serious games in the domain of crisis management. We hope we haven't missed some essential ones. If so, we would be very grateful to the reviewer to suggest the specific references that he/she thinks is still necessary to include.

C2 : ANYCaRE has to be compared with existing crisis management RPGs, in order to highlight its specificity and its interest.

R2: As suggested we highlighted the specificity and interests of ANYCARE games compared to others with similar purpose, by adding the following paragraph (p6): "If ANYCARE game falls in the most common category of "face-to-face multi-player experience with lively interactions between players, it is one of the few role-playing game dealing with early warning systems (EWS) and allowing to test new forecasting products (Solinska-Nowak et al., 2018). The UK Met office developed the Forecast-Based Early Action Game simulating a community affected by floods who need to prioritise their actions based on severe weather forecast (UK Met Office, Deltares, & Red Cross Red Crescent Climate Centre, 2018). Another role-playing game called Adapted Technologies for Early Warning Systems: Playing with Uncertainty aims to illustrate the complexity of decision making on EWS and the need of integrating all its components by playing with different scenarios of economic development, stakeholders' participation, technology and uncertainty (Garcia & Fearnley, 2018)."

C3 : The rules of the game are not clearly presented in the paper, limiting the capacity of the authors to explain and justify the game design choices.

R3: We made an effort to clarify the presentation of the rules and justify the game

design choices by redistributing the information provided in more appropriate sections. The pieces of text describing the rules of the game that was initially in section 2.1.2. were moved to section 2.2.3. The objective of the simulation was also clarified in section 2.2.2. (p12).

C4 : The game is intended for a wide and varied audience (from people with no knowledge of the subject to experts). Is it possible to consider all these audiences with the same game? How does this impact game design? How is managed the difference of expertise/knowledge between players?

R4: We manage the difference of expertise (specially the lack of expertise in forecasting products) by dedicating a little more time to the learning/description of the products at the beginning of the game. If necessary, one of the game organizer, more skilled in forecasting products, may also facilitate the process by helping the Âń forecasters Âż to interpret the model's outputs. In terms of the game outputs, it is very interesting to see that having at least one player with a crisis management experience really helps highlighting the complexity of the decision-making process as one decision may have many imbricated consequences that does not easily come to mind when you haven't faced them before.

Detailed comments from reviewer #1

Page 3: L5: "The simulations are built based on the hypotheses that dynamic (near)-real-time impact information." What is the meaning of near here?

R1: We agree with the reviewer that this word is more confusing than helping the understanding and we therefore deleted it.

L9: "Communicate more targeted warnings and emergency guidance messages to help the public understanding..." Specify the type of aimed audience.

R2: Here we are talking about the general public and more specifically to the persons exposed to the threat that the warnings are expected to alert for. We added the term

Âń exposed Âż to specify the type of public we are talking about.

L13: "The game design allows the player..." How?

R3: to answer this question, the sentence as be changed for: Âń By first providing the players with hazard-forecast information alone and then adding impact-based forecasts at each round of the game, it allows them to progressively integrate the use of impact-based models' outputs and reflect on the usefulness of such information in supporting the collective emergency decision-making process. Âż

L15: "Through (semi-) realistic "what if" scenarios..." What is the meaning of semi here?

R4: We agree with the reviewer that this word was not needed we therefore deleted it.

Page 5: L3: "The learning outcome of ANYCaRE is the improved weather risk-related decision-making for emergency response through modern multi-source inputs." Explain what is a modern multi-source inputs; give some examples.

R5: The following additional sentence with examples was added (p7): "Thus, participants get informed for new forecast products (like multi-hazards or impact-based model outputs) and informational systems (also involving crowd-sourced information), and then practice their understanding and interpretation of those tools through playing a realistic crisis scenario Âż

L9: "By observing players' debate and decision-making processes during the simulations the experiment aims ultimately to obtain first conclusions on "if "and "how" improved multi-model (and potentially multi- hazard) outputs, including information on impact-based vulnerability data, can support the decision chain in European warning systems towards better responses." Decompose the sentence and give more explanation on this point.

R6: As suggested, we decomposed the sentence and added extra details on the methods used (p7): "During the game, at least one of the game developer act as an "observer" of the discussions and debate guiding the decision-making. In addition, at the end of the game, a specific time is dedicated to debrief about the experience. With the use of such qualitative methods the experiment aims at assessing "if "and "how" improved multi-model (and potentially multi-hazard) outputs, including information on impact-based vulnerability data, can support the decision chain in European warning systems towards better responses. Âż

Page 6: L1: Table 1 requires more explanation in the text.

R7: As suggested, we reframed the sentence to be more explicit and added another sentence giving examples on how this table can help future play-game developers to support their design choices (p8-9): "Table 1 describes the key components (storyline, roles, simulations) necessary for the definition of the scenarios and summarizes the questions that guided our design choices and further reflexions we considered for each of those components. [. . .] For instance, it could help developers to consider options with respect to choices of real world versus virtual territory or time frame and pace of the decision-making related the type of hazard considered. Âż

L16: "For example, in ANYCaRE game the GMs is responsible, among others, to provide feedback to the players about the hydro-meteorological observations highlighting the relevant safety decisions that should have be taken at each playing round (similar to a weather reporter)." What is the basis for the information that the GM proposes?

R8: The weather-related scenario we propose in the game is made from a selection of real past events from different places, which according to our expert judgment are either providing very clear signal that the event will or won't happen, or an unclear signal which highlight the necessity of additional information (e.g. social media, higher resolution/local prediction) in order to envisage the possible development of the situation. The whole scenario (data and actions to be taken) is simplified enough to fit in the controlled environment of the game. It is important to mention that crisis management is a wicked problem and there is no "right" action. The idea of the game is to illustrate the complexities of taking emergency decisions when the available information is highly uncertain. One of the main purposes of the game is to teach/show the necessity to always evaluate possible action considering its cost/benefit in relation to the likelihood of the occurrence. Therefore, to assess the relevance of the actions the players selected on D-1 for the D-day, we use (hydrological or meteorological) observations for the D-day. This information allows drawing conclusion on whether the event happened or not as forecasted to confront it with the decisions made by the group of players.

Page 7: L9: "A panel of roles' description to be distributed among the players (randomly or based on their real-life expertise)". This choice of role is important and could be a part of the training/rising awareness strategy; it would be interesting to discuss this point in the paper.

R9: It is right that this point is important and 2 sentences were added to discuss it further (P10-11): "Some roles, as the ones of the forecasters or the one of the group leader, may be distributed carefully as they require either a strong expertise or leadership. It is clear that the way those roles are played may influence a lot the results of the decision-making process and confidence of the overall group in their collective decisions. Nevertheless, a random distribution of the roles may also raise the participants' awareness about the capabilities and level of expertise necessary to interpret and act upon such complex, dynamic and uncertain crisis situation. Âż

L25: "A Three rounds of decision-making are played successively to simulate the progression of the hazard from its early detection to its landfall." Is it always 3 rounds (whatever the application)? Can you justify this choice?

R10: In fact, the choice of 3 rounds is a trade off between realistic representation of the evolving hydrometeorological facts handled in the scenario and the practicability of the game exercise. The following justification was added to support our choice of three rounds (p12 L3-7). "By using multiple rounds we allow the players to experience evolving hydro-meteorological facets and test different decision-support tools, which

give more and more accurate information, as it gets closer to the event occurrence. The repetition of the decision-making process over several rounds also helps the players getting better at managing their roles and learning from practice.

Page 8: L3: "Based on the selected emergency activity, the group further agrees if they would provide some generalized advice for safety to the public (e.g., "If inside, move to higher floors", "Be prepared for electricity disruptions") or if they would proceed to more detailed emergency orders in specific area(s) of the territory (e.g., "Evacuate immediately")." Can you give more details on this process?

R11: To make it clearer that the type of decisions they need to take collectively are based on a list of options proposed in the decision sheet the emergency leader has to complete for each trial of each round, we added a reference to the Annex A which display the decision sheet provided to the players (p13 L21). If none of the options convince them they can tune them to their needs.

L15: "Players get a few minutes to become familiar with their role and to introduce it to the rest of the group before the main game simulations start." Is it sufficient (notably for players without expertise in the field)?

R12: Thank you for raising this delicate point. In some cases we have noticed that the players tend to feel the preparation time is too short, especially for the ones who play the roles of forecasters who have several model outputs to analyze and comment. To help the process when the players are not that experienced or familiar with the field, a little more time is dedicated to this learning phase during round 1 of the game and one of the game organizer is facilitating the process by training the Âń forecasters Âż to interpret the model's outputs. However, the reason why we kept the preparation phase short is that, in the contexts of our past experiments, we had to limit the game-play to 1h30 (2h with debriefing). In future experiments, if more time is available, the timing could be adapted or trainings to new forecasting products could be envisaged as a separate preparation before the game is played. Nevertheless, as time pressure is

also an important element of emergency decision-making we are attentive to limit the time for interpreting, sharing information and deciding to make sure players experience the type of pressure happening is real crisis situations.

L22: "Following this first step, the players have to select (among some pre-established options) the best way to communicate their decisions to the targeted public." How many options? What freedom is left to the players? Can they suggest new options?

R13: All the four available communication options are written in the decision worksheet provided in Annex A (a reference to this document was added in the text (p12 L12). It would be very interesting to let more freedom to choose how and what to communicate to the public, nevertheless in the limited time available (1h30 for the game-play) and the current setting of the game-play, it was not possible. A way to improve this aspect in the future might be to create a new role of "communication officers", physically separated from the decision-maker group that would be in charge of selecting communication means and writing the appropriate messages to the general public.

Page 9: L12: "Each round is composed of two trials, one where only existing basic hydro-meteorological forecasts are available and the second where additional more sophisticated decision-support products are provided." The interest of these two trials is only explained later in the text. You should introduce the justification of this game design choice here.

R14: The specific justification of these 2 trials is given for each of the 2 scenarios (Flood and strong winds) described in section 3. It goes into much details related to the specific input data and simulation for each of the scenario. Therefore we think this would not be relevant to transfer all this information in section 2.2.3, where only the information common to both scenarios is provided.

L18: "This time includes 1-2 minutes for the GM to present a short summary of the (hydro-) meteorological situation and consequences of 20 the decisions that have been taken at the previous round." How are evaluated/calculated these consequences?

R15: In this version of the game no detailed calculation of the consequences was made. Consequences of the decisions were deemed relevant when protective actions were put in place before the actual occurrence of the event, and deemed irrelevant when no or insufficient actions were put in place at the time the event occurred. The actual time of occurrence of the event was deducted from (hydro-) meteorological observations (see response R8 for more details).

Page 11: L1: "In each round, the players receive area-specific information so that they could make distinct safety choices adapted to the predicted hazard in each area of Anywhere City." Why using different areas? What is the impact on the game?

R16: In the flood scenario, the 3 areas are not prone to the same flood dynamics, as the smaller catchment in area A is subject to flash floods while other catchments' responses may be more predictable and leave more time for protective actions. The use of different areas aims at exploring decision dilemmas related to connectivity and communication issues as well as spatial and temporal heterogeneity of the hazard characteristics. For instance, sometimes it is more pertinent to close all the schools and cancel school transportation than just the ones schools that might be directly impacted. Not only this decision is less confusing to communicate to the public but also it takes into account that flooding might impact school transportation even though schools are not threatened. Nevertheless, in case of very localized threat (like flash floods), limiting protective actions to the area specifically exposed also makes sense, as this measure is only disruptive to the part of the population that is really at risk.

Page 13: L17: "In particular, in the second trial meteorologists receive model calculations outputs from the "Expert System for Consequence Analysis and Preparing for Emergencies" (ESCAPE) model calculations, and interpret them for the use of civil protection." Are these results pre-calculated or are there live simulations made according to the players' choice? In this case, what is the duration of a simulation?

R17: For practical reasons and to limit the duration of the simulation, the results are

pre-calculated with the specific wind direction and wind speed to simulate the prevailing conditions before the Asta-thunderstorm. (The correct wind speed and wind direction was collected from the weather observation archive of Finnish Meteorological Institute and inserted to ESCAPE model to create the dispersion calculation.) Nevertheless, it would be interesting to adapt the game to try live simulation in the future, since it would be possible with ESCAPE tool.

Page 15: L17: "To trigger debate rather than winning spirit in the performed experiments, no specific penalties were assigned to the options listed in the worksheets. Instead, in the first round the GMs highlighted the three-fold common goal of the emergency managers to: i) insure citizens' safety and prevent loss of life; ii) prevent disturbances in social life, which make people unhappy and reduce wellness; iii) minimize their expenses for protective measures compared to the actual needs." It is not clear. Can you reformulate/explain this point?

R18: We also believe this paragraph doesn't make sense in this section and is not bringing anything to the understanding of the game so we chose to delete it.

Page 16: L12: "Players noted their appreciation for the 'stimulating' (6 instances), 'very fun' (4 instances) learning approach experienced in ANYCaRE." On how many instance? Was there also criticism from the players? Why did you choose this type of evaluation rather than an anonymous questionnaire?

R19: The instances are counts of the number of comments that enter these categories. The details and total number of instances are given in table 4. To be more explicit in the paper, the sentence has been modified (p23) for: Âń Based on the categorization of the comments provided on post-it notes during the debriefing sessions, we found that players noted their appreciation for the 'stimulating' (6 instances), 'very fun' (4 instances) learning approach experienced in ANYCaRE (see table 4 for details of all comments and related instances). Âż Critics are listed in the right column of table 4 with the title Âń improvements proposed... Âż. The debriefing sessions combine the

richness of qualitative and open questions and the anonymity as players were proposed to use anonymous post-it notes that could be commented collectively.

Page 17: L12: "Although, differences between Trial 1 and Trial 2 were not always obvious in terms of the selected emergency activity on the worksheets, in most of the cases, players in the four experiments rated their confidence higher when they passed to Trial 2." Trials 1 and 2 are combined in the same game session. Does that not introduce a bias in the comparison of the two trials? Shouldn't it have been better to make two independent experiments, one with classical information (i.e. trial 1) and the other with advanced information (trial 2)? The variability between the groups are not discussed; it would be an interesting point.

R20: When we designed the game, we examined this option but the reasons why we choose to combine the 2 trials in the same game are the following: - Trails 2 provide additional impact-based information but hazard-forecast information is the basis, so if we would separate the 2 trials, we would still have to repeat the information provided in trail 1. We believe this repetition would take more time and would make the exercise more repetitive and tedious. - The scenario is the same for Trial 1 and 2 only the available information changes. If players would have to play the same scenario twice, knowing the end of the story from the first play would anyway influence their decisions in the second play.

Page 19: L4: "In summary, the respondents indicated the following take away messages..." It would have been interesting to compare learning through play with that obtained with traditional methods (e.g. courses).

R21: The settings of the past ANYCARE experiments didn't allow proposing this option, but we agree that this would definitely be an interesting target for future research using ANYCARE.

L23: "To facilitate the representation and relevant intervention of all the roles and their particularities in the debate, it was suggested that very detailed responsibilities and

constrains should be assigned to each role at the beginning of the game; allowing the players to set up a common strategy with the other persons playing the same role (8 instances)." Another way to avoid the 'leader effect' is to give each player only part of the information, forcing them to communicate to build a common strategy.

R22: We actually did this especially in the strong wind simulation. We gave all of the roles slightly different information and tools (tools that those specific roles would use in their work) and encouraged them to communicate further to the other roles the information they had in their materials. (The material of each role included guiding questions or recommendations, like: "What can you see in this graph?" "Please, communicate the relevant information to the other roles."

Page 21: L23: "Adjustments of the current flood game are already considered to prepare experiments for Italian pupils (e.g., between the age of seven to thirteen years old) with the objective of raising awareness on i) the crisis decision-making process and the challenge of school-related decisions, and ii) the appropriate behaviours of students and their parents in case of flash flooding." How?

R23: As this action is only at his early stage, we believe there is no point of giving this information in this paper we therefore prefer to delete this sentence.

Please also note the supplement to this comment:
https://www.nat-hazards-earth-syst-sci-discuss.net/nhess-2018-244/nhess-2018-244-AC1-supplement.pdf

**Supplement:**

[revised manuscript text omitted]

**(a)**

| | |
|---|---|
| **A** | ▪ Residential area (e.g., 1,000 residents) ▪ Campsite in flood zone |
| **B** | ▪ Main urban area (e.g., 100,000 citizens) ▪ Schools in flood zone |
| **C** | ▪ AnyDay Festival area (e.g., 10,000 participants) ▪ Main bridge in flood zone |

**Anywhere City**

Flood Scenario

**(b)**

| | |
|---|---|
| **A** | ▪ Urban and residential area (e.g., 50,000 inhabitants) ▪ Jurassic Rock Festival area (e.g., 25,000 participants) |
| **B** | ▪ Sulkavan Lake - Sulkavan Soutu Boatrace (e.g., 4000 boatment) |
| **C** | ▪ Main road connecting Sulkava and Juva (area prone to tree falls) |

**South Savo Municipality**

Thunderstorm Scenario

**Figure 2: Presentation and brief description of the territory considered in the storyline of ANYCaRE for the: (a) Flood scenario and (b) Strong wind scenario. Each area of the territories includes attributes for special consideration in the emergency decision-making (e.g., camping, schools, dangerous intersections, industries) representing critical points for intervention in European cities.**

[Figure]

**Figure 3: Schematic illustration of the gaming timeline (rounds 1 to 3) and the information provided to the players of ANYCaRE for the: (a) Flood scenario and (b) Strong wind scenario. Each of the three game rounds (R1, R2, R3) played in the experiments corresponds to a daily or hourly time step before the festivals that, according to the storylines, are held on Saturday. In the second trial of each round, the players receive additional decision-support tools including high-resolution forecasts and impact-based vulnerability inputs.**

[Figure]

**Figure 4: From (a) to (d): Players' debate during the role-playing simulations of ANYCaRE in: (a) Helsinki Experiment (21 September, 2017); (b) Grenoble Experiment (24 January, 2018); (c) UGA Experiment (11 April, 2018); (d) Mikkeli Experiment (17 May, 2018). (e) Example of players' post-experiment debriefing post-it notes (photo taken in Helsinki Experiment).**

---

## Author Comment (AC2) · 25 Feb 2019

Comments from reviewer #2 I think this paper is very interesting and is an important addition to the literature. It well describe the "ANYCaRE" role-playing game to simulate crisis decision-making and management arousing curiosity and making me want to play and deepen the sub- ject deal with. After reviewing it, I would like to experience it myself as a player tester. The authors are able to explain and justify the game design choices in a detailed and punctual way rich in useful particular to understand the consequences of one decision rather than another on crisis management. ANYCaRE is presented as a serious games particularly useful to educate future crisis managers,

as well as to spread a culture of risk also for general public promoting awareness and correct behaviors which are fundamental for public safety. A very precious tool for all (from citizens with no knowledge of the subject to experts). I believe the importance of this paper stems from the applicability of the approach to the several, but I have some reservations about a so difference players' scientific background. My suggestions are essentially that authors should also calibrate the game to inexperienced users, simplifying in this case the data sets (considered by someone too difficult to interpret). The adequacy of the development of the project described, as it is currently stands, shows the authors' analysis is both rigorous and generalizable. A very well written interesting manuscript, pleasant to read and fluent: a beautiful synthesis of a role-playing game with the education intent our Society needs that can be broadly and usefully applied.

Response: We are very grateful to the reviewer for her thoughtful and appreciative comments. We agree that specifically the forecast information is quite complex to interpret for inexperienced users and it is why we tend to manage the difference of expertise by dedicating a little more time to the learning/description of the products at the beginning of the game. If necessary, one of the game organizer, more skilled in forecasting products, may also facilitate the process by helping the Âń forecasters Âż to interpret the model's outputs. Nevertheless, our intention when conducting future simulations is to better adapt the forecasting information to the audience, for instance using products that are commonly used in the country. Only the new impact-based forecasting products and social media application might still be new to the players but it is in fact the understandability, efficacy and transferability of such products that we intend to test with ANYCARE simulations.